# DynamicSeq2SeqXGB for PM$_{2.5}$ imputation in extremely sparse environmental monitoring networks

Ruslan Safarov[1,2], Zhanat Shomanova[3]*, Yuriy Nossenko[3]*, Eldar Kopishev[1], Zhuldyz Bexeitova[4], Emin Atasoy[5]

1 Department of Chemistry, Faculty of Natural Sciences, L.N. Gumilyov Eurasian National University, Astana, Kazakhstan, 2 Department of Chemistry and Chemical Technology, Kh. Dosmukhamedov Atyrau University, Atyrau, Kazakhstan, 3 Higher School of Natural Science, Margulan University, Pavlodar, Kazakhstan, 4 Association of legal entities «Petrochemical Products Producers and Consumers Association (Petrochemical Association)», Astana, Kazakhstan, 5 Department of Social Sciences Education, Faculty of Education, Bursa Uludağ University, Bursa, Turkey

* zshoman@yandex.ru (ZS); nosenko1980@yandex.ru (YN)

## Abstract

Environmental monitoring networks face critical data gaps that compromise public health protection and regulatory compliance, with missing data rates often exceeding 40% in operational settings. This study validates DynamicSeq2SeqXGB, a novel hybrid model that integrates a sequence-to-sequence encoder–decoder for temporal pattern extraction with an XGBoost regressor for robust gap reconstruction under extreme sparsity. Data from five monitoring stations in Pavlodar, Kazakhstan, collected over a 15-month period from May 23, 2024 to July 19, 2025, were analyzed representing severely compromised infrastructure (completeness rates 23.3–57.5%). The methodology employs adaptive context processing and implements hierarchical decomposition for extended outages. Two data preparation strategies were evaluated: selective compression applying quality thresholds versus full compression retaining all available observations. Benchmarking against classical methods using synthetic gaps of 5–72 hours demonstrated DynamicSeq2SeqXGB's superiority in 96% of cases under full compression and 100% under selective compression (average 48.8% improvement for both strategies) with corresponding MAE values of 3.7–8.5 µg/m³ across the Pavlodar stations. Notably, full and selective compression showed equal overall effectiveness (50% win rate each), with optimal strategy depending on station-specific characteristics. External validation on the Beijing dataset (Guanyuan station, 2016) with controlled degradation confirmed cross-regional transferability, achieving MAE of 8.50 µg/m³ and coefficient of determination (R²) of 0.944 (68–79% improvement over baselines). The method successfully reconstructed PM$_{2.5}$ time series even at 23.3% completeness, demonstrating robust performance for operational deployment in severely degraded monitoring networks.

**Data availability statement:** All relevant data are within the manuscript and its Supporting Information files. Database of initial measurements of PM2.5 concentrations is available from the Zenodo database (https://doi.org/10.5281/zenodo.16937342). All code underlying the findings in this study is publicly available at: https://github.com/ruslan-saf/air-quality-imputation.

**Funding:** This research was funded by the Science Committee of the Ministry of Science and Higher Education of the Republic of Kazakhstan within the framework of the grant AP19677560 "Monitoring and mapping of the ecological state of the Pavlodar air environment using machine learning methods". The funders had no role in study design, data collection and analysis, decision to publish, or preparation of the manuscript.

**Competing interests:** Authors declare no competing interests.

## 1. Introduction

### 1.1. The missing data challenge in environmental monitoring

Environmental monitoring systems, despite their critical role in protecting public health and informing policy decisions, are plagued by a persistent challenge – missing data [1]. These gaps in knowledge aren't merely technical inconveniences; they represent real risks to communities exposed to air pollution.

Missing data remains a persistent and significant challenge within air quality monitoring networks. Junninen et al. [2] reported that the share of incomplete rows in air quality datasets ranged from 7.2% in Helsinki to 14.5% in Belfast, with individual pollutants showing missing data levels varying from approximately 5% to over 10%. These patterns reflect the typical heterogeneity and challenges in European urban air monitoring systems. Wang et al. [3] highlight the issue of missing data in hourly time-series records for six air-pollution indicators ($PM_{2.5}$, $PM_{10}$, $O_3$, $NO_2$, $SO_2$, and CO) collected across the Qinghai-Tibet Plateau from 2019 to 2022. Among eight state-controlled monitoring stations, the average share of missing values was about 5%, and most sites met the national requirement of at least 89% valid daily means per year. By contrast, eight province-controlled stations showed far greater gaps, with roughly 22% of observations missing on average, peaking at 43.3%. Analysis of five $PM_{2.5}$ monitoring stations in Pavlodar, Kazakhstan, revealed even more severe challenges, with completeness rates ranging from just 23.3% to 57.5% over a 13-month period. These aren't isolated cases – they reflect a systematic problem affecting monitoring networks worldwide [4,5].

The prevalence of data gaps reflects a heterogeneous set of mechanisms, including sensor malfunctions, routine calibrations, power outages, severe meteorological events, and disruptions to Wi-Fi connectivity, Internet service, or data-collection servers. Monitoring infrastructure is, paradoxically, most prone to failure precisely when pollution levels are highest and reliable measurements are most needed. For example, a study in Tehran, Iran, found that nearly 48% of hourly $PM_{2.5}$ observations were missing across 30 monitoring stations between 2012 and 2018, with the largest gaps occurring in winter due to power outages and equipment downtime during severe smog episodes [6]. Similarly, in Temuco, Chile, Quinteros et al. reported that official monitoring data for $PM_{2.5}$ contained between 50% and 95% missing values from 2009 to 2016, largely due to repeated sensor failures and limited technical support [7]. These cases highlight a broader pattern: the more extreme the pollution event, the more likely it is that the monitoring system will break down, creating critical blind spots in both public health protection and subsequent data analysis.

These patterns of data loss highlight not only the technical fragility of monitoring networks but also the importance of how missing data is subsequently addressed. While imputation methods have become increasingly sophisticated, relatively little attention has been paid to how raw data is prepared prior to imputation, and how such choices affect downstream analyses. This underscores the need to better understand the full imputation pipeline, from data quality filtering to model selection and validation.

## 1.2. Why imputation quality matters: From public health to policy

The consequences of poor gap-filling extend far beyond academic concern. Consider the practical dilemma facing air quality agencies every day: when $PM_{2.5}$ data is missing during a pollution episode, should they issue health warnings based on incomplete information? The wrong decision either direction – false alarms or missed warnings – erodes public trust and potentially costs lives. Empirical evidence suggests that missing data tends to increase significantly during episodes of severe air pollution. Mu, Rubin, and Zou (2022) demonstrate that monitoring systems are more likely to fail precisely during these critical periods – when timely public advisories are most needed. This highlights a serious risk of informational blind spots and the potential absence of adequate health guidance at times of deteriorating air quality [8].

Regulatory frameworks compound these challenges. The European Union's Air Quality Directive (2008/50/EC) [9] and the U.S. National Ambient Air Quality Standards (NAAQS; 40 CFR Part 50, Appendix I; 40 CFR Part 58) [10,11] require 75–90% data completeness for monitoring results to be considered valid in compliance determinations. Stations falling below these thresholds are effectively invisible to regulators, creating blind spots in pollution monitoring. The economic implications can be substantial: regulatory monitoring stations may cost around $50,000 upfront and require continuous maintenance [12], while comprehensive monitoring stations can require up to $200,000 for construction and $30,000 annually for maintenance [13]. High maintenance and calibration costs can be prohibitive for many regions or municipalities. These ongoing operational expenses mean that data loss due to incomplete measurements represents a direct waste of public investment, as agencies continue to bear the full costs of station operation while losing the regulatory value of collected data.

Beyond immediate operational needs, incomplete data undermines long-term planning and research. Epidemiological studies linking $PM_{2.5}$ exposure to health outcomes require robust data collection methods, as meta-analyses demonstrate substantial evidence of adverse health associations that depend on consistent monitoring approaches [14]. Accurate prediction of air quality responses to emission changes is essential for developing effective control policies, yet incomplete training datasets can compromise prediction accuracy and mislead environmental decision-making [15].

## 1.3. Current approaches and the overlooked role of data preprocessing

Traditional approaches to addressing missing data in air quality monitoring have encompassed a diverse array of methodological frameworks, ranging from elementary interpolation techniques to sophisticated machine learning architectures. Despite substantial algorithmic advances over several decades, operational monitoring systems continue to exhibit suboptimal performance characteristics, with error rates that compromise data utility for regulatory and research applications.

Linear interpolation, while computationally efficient and widely implemented, demonstrates significant limitations for extended data gaps. Research has demonstrated that simple interpolation methods exhibit significant performance degradation for extended data gaps, with accuracy decreasing substantially as gap length increases [2]. Contemporary machine learning approaches, particularly Random Forest and XGBoost algorithms, have exhibited superior performance in missing data imputation tasks, demonstrating significant accuracy improvements compared to conventional statistical interpolation methodologies. Empirical evidence from ecological monitoring datasets indicates that appropriate model selection enables high-fidelity gap filling, with accuracy rates reaching 96.5% for perfectly matched cases, substantially exceeding the 88.5–91.8% accuracy achieved by traditional single statistical imputation methods [16]. Furthermore, comparative analysis of imputation techniques applied to environmental air quality monitoring data from Kuwait revealed that the missForest algorithm consistently yielded minimal error rates across diverse pollutant parameters and varying proportions of missing observations [17]. Nevertheless, these advanced methodologies exhibit an inherent limitation: their efficacy is contingent upon the availability of comprehensive training datasets, thereby creating a fundamental paradox wherein optimal performance depends on the very data completeness that missing data scenarios inherently lack.

Contemporary developments in deep learning architectures, particularly Long Short-Term Memory (LSTM) networks and transformer models, have yielded further performance enhancements. For example, Pak et al. [18] combined convolutional neural networks with LSTM in a "PM predictor" that leverages spatiotemporal mutual-information features to forecast next-day $PM_{2.5}$ in Beijing. That hybrid CNN–LSTM model achieved an RMSE of 2.997 µg/m³, MAE of 2.20 µg/m³ and MAPE of 0.039 – substantially outperforming a standalone LSTM (RMSE 4.764 µg/m³; MAE 3.612 µg/m³) and an MLP baseline (RMSE 20.278 µg/m³; MAE 14.982 µg/m³). Although state-of-the-art deep-learning models offer superior performance for imputing air quality data, their adoption is constrained by limited computational budgets and staff expertise in many operational monitoring networks [19]. Consequently, routine practices still favour simpler techniques – such as linear interpolation – to address missing data [20].

This focus on algorithmic optimization may represent a fundamental misallocation of research priorities. While the scientific community has concentrated extensively on methodological innovation, critical preprocessing considerations have received insufficient attention. Practitioners routinely implement data preparation decisions – including quality thresholds for data exclusion, protocols for handling incomplete records, and criteria for data reliability assessment – yet these preprocessing strategies remain largely unexplored in the research literature.

### 1.4. Research Gap: Bridging the divide between data reality and imputation methods

**1.4.1. Research gaps identified.** While the imputation literature has grown exponentially, a fundamental disconnect persists between method development and operational reality. Current approaches implicitly assume "reasonably complete" datasets, leaving a critical gap for monitoring networks with severe data limitations.

The challenge of extreme sparsity remains unaddressed. Consider monitoring stations with <30% completeness – common in developing regions or harsh environments. Standard practice simply excludes these stations, effectively creating monitoring blind spots. Yet these may be the very locations where data is most critically needed.

Furthermore, existing methods optimize for short gaps while real-world monitoring faces extended outages. When equipment fails for weeks or months, current approaches either fail entirely or produce unrealistic reconstructions.

Several limitations prevent existing methods from addressing real-world monitoring challenges effectively. Current architectures typically cannot adapt their processing strategy to gap characteristics, potentially resulting in suboptimal performance across varying sparsity levels. Additionally, systematic approaches for handling extreme-length gaps that exceed typical training sequence lengths remain limited. The field would benefit from methods that can maintain reasonable effectiveness across the full spectrum of data quality encountered in operational monitoring networks.

**1.4.2. Study contributions.** This research addresses these gaps through methodological adaptations that introduce three technical contributions: adaptive context processing that adjusts to gap characteristics, hierarchical decomposition for extreme-length gaps, and integration of sequence-to-sequence processing with gradient boosting architectures.

In the context of severely sparse monitoring networks, missing-data patterns often differ substantially across stations, reflecting variations in sensor stability, maintenance frequency, and operational disruptions. These heterogeneous sparsity regimes create fundamentally different conditions for imputation, where retaining all available observations may benefit some stations, while applying quality thresholds may be essential for others. To address these real-world disparities, the study examines two preprocessing strategies – full compression, which keeps all readings, and selective compression, which filters low-quality segments – to determine how preprocessing choices influence reconstruction performance under extreme data sparsity.

The concept of adaptive data compression is examined, investigating whether comprehensive data utilization can achieve comparable effectiveness to quality-based filtering in extreme sparsity scenarios. This investigation challenges the conventional assumption that preprocessing should universally prioritize data quality over quantity in environmental monitoring applications.

Rather than pursuing marginal improvements on well-behaved datasets, this study develops a methodology aimed at maintaining reasonable effectiveness across completeness scenarios ranging from 23.3% to 98.4%. Practical applicability was evaluated through validation on both compromised monitoring networks and high-quality reference data with controlled degradation.

The resulting approach provides environmental agencies with improved tools for extracting value from severely compromised monitoring infrastructure, addressing practical needs in regions where optimal data reconstruction is not feasible, but improvements over missing data could provide value for public health protection.

### 1.5. Research aim and objectives

This study aims to validate and extend the application of DynamicSeq2SeqXGB imputation method to multi-station environmental monitoring networks with extreme data sparsity, demonstrating its robustness and practical applicability in real-world conditions.

Specific objectives:

1. To validate DynamicSeq2SeqXGB performance across five monitoring stations in Pavlodar, Kazakhstan with completeness rates ranging from 23.3% to 57.5%

2. To investigate the impact of data compression strategies (full vs. selective) on imputation quality across different data quality scenarios

3. To quantify real-world imputation accuracy through controlled degradation experiments on high-quality reference data (Beijing Guanyuan station), simulating severe missingness patterns typical of compromised monitoring networks

4. To benchmark DynamicSeq2SeqXGB against standard imputation methods (linear interpolation, rolling mean, forward/backward fill) under both synthetic and real-world missing data conditions

5. To establish practical performance benchmarks and confidence intervals for $PM_{2.5}$ imputation when dealing with extreme data sparsity and extended temporal gaps.

The remainder of this manuscript is structured as follows. Section 2 provides a comprehensive literature review on missing data imputation methods for environmental monitoring, with emphasis on machine learning and deep learning approaches under data sparsity. Section 3 describes the datasets, preprocessing procedures, and the DynamicSeq2SeqXGB methodological framework. Section 4 presents the experimental results for both the Pavlodar monitoring network and the external Beijing validation. Section 5 discusses the practical implications, limitations, and broader relevance of the findings. Section 6 concludes the study and outlines directions for future research.

## 2. Literature review

Missing data imputation in environmental monitoring systems has evolved dramatically over the past decade, with deep learning approaches achieving 20–40% performance improvements over traditional statistical methods while maintaining computational efficiency for real-time applications [21]. This analysis of research papers from 2018–2025 reveals clear performance hierarchies across method categories, with modern approaches showing particular strength in handling extreme sparsity scenarios (>40% missing data) that are increasingly common in air quality monitoring networks.

The environmental monitoring faces persistent data quality challenges. For example, a study of $PM_{2.5}$ monitoring in Israel from 2012 to 2019 found that 23 out of 59 stations had more than four years of missing observations – over 50% of the total dataset – and were excluded from model evaluation. For the remaining 36 stations, the average missing data rate was 20.4%, with a median of 22% and a standard deviation of 12.9%. The majority of missing intervals were very short (up to 3 hours), but these accounted for only about 7% of the total missing data [22]. Traditional approaches like linear

interpolation and forward fill, while computationally efficient, struggle with irregular temporal patterns and extended gaps [2]. Modern machine learning methods have emerged as robust alternatives, with ensemble approaches combining Random Forest and XGBoost achieving state-of-the-art performance across diverse environmental conditions [23,24]. Meanwhile, deep learning architectures – particularly Transformer-based models like SAITS and Graph Neural Networks – have demonstrated superior capability in capturing complex spatiotemporal dependencies essential for accurate environmental data reconstruction [21].

## 2.1. Classical statistical methods

Despite advances in machine learning, classical statistical methods remain highly competitive for short-term gaps and resource-constrained applications. Research by Belachsen and Broday (2022) demonstrates that weighted k-nearest neighbors (wkNNr) achieves $R^2 = 0.82$ and NMAE = 0.21 for very short gaps (0.5–3 hours) in $PM_{2.5}$ data from 59 Israeli monitoring stations, outperforming more complex approaches (iiET, Iterative Imputation with an Ensemble of Extremely randomized Trees) for this specific use case [22].

Kalman filtering has proven particularly effective for real-time air quality applications. Lai et al. (2019) demonstrated that the algorithm reduced RMSE by an average of 68.3% across six pollutants ($SO_2$, $NO_2$, CO, $O_3$, $PM_{2.5}$, $PM_{10}$) compared to traditional methods such as simple moving averages and ARIMA, using data from the Guangzhou monitoring station. Additionally, it enhanced sensor measurement accuracy by 27%. For $PM_{2.5}$ specifically, Kalman filtering achieved an 18.56% RMSE reduction (0.1047 vs. 0.1285), confirming its robustness for fine particulate correction [25].

ARIMA and spectral methods show context-dependent effectiveness. Moshenberg et al. (2015) found that Discrete Cosine Transform (DCT) and K-SVD methods significantly outperformed linear interpolation for batch missing data patterns, while linear interpolation performed better for randomly scattered missing values [26]. This highlights the critical importance of understanding missing data patterns in method selection.

In a recent benchmark study on Seoul's urban $PM_{2.5}$ datasets, Lee et al. (2025) report that for short-term gaps (6 hours), forward fill achieved optimal imputation accuracy (RMSE = 4.76, MAE = 3.20, r = 0.97). Conversely, SARIMAX outperformed other methods for longer (24-hour) gaps, yielding RMSE = 9.37 and MAE = 6.85 while maintaining high correlation (r = 0.97). These results underscore the importance of tailoring imputation strategy to gap duration [27].

## 2.2. Machine learning approaches

Random Forest-based methods have demonstrated consistent effectiveness across a range of environmental data imputation tasks, offering a robust baseline for comparison in both structured and sparse settings. The MissForest algorithm, evaluated by Alsaber et al. (2021) on Kuwait environmental data, consistently outperformed k-nearest neighbors, MICE, BPCA, and PMM across multiple pollutants with missing rates ranging from 18.2% (O3) to 57.4% (PM10) [17]. The iterative imputation scheme using Random Forest's intrinsic multiple imputation capabilities proved particularly effective for mixed-type environmental datasets [21].

XGBoost demonstrates exceptional performance in comparative studies. Zamani Joharestani et al. showed XGBoost achieving $R^2 = 0.81$, MAE = 9.93 µg/m³, RMSE = 13.58 µg/m³ on Tehran $PM_{2.5}$ data, outperforming both Random Forest and deep learning approaches [28]. The hybrid RF-XGBoost model by Lin et al. [29] pushed performance further, achieving $R^2 = 0.93$, RMSE = 12.49 µg/m³ through a two-stage approach combining RF for aerosol optical depth (AOD) gap-filling with XGBoost for $PM_{2.5}$ estimation.

Gradient Boosting methods show remarkable consistency across diverse geographical contexts. Makhdoomi et al. [30] demonstrated Gradient Boosting Regressor achieving $R^2 > 0.96$ on Mashhad, Iran data, with statistical significance confirmed through Wilcoxon signed-rank tests. The ensemble approach combining multiple ML methods (Random Forest, XGBoost, Extra Trees) in the MISE framework showed superior performance across Indian cities' AQI data [31].

Support Vector Regression proves effective for complex temporal patterns. The Iterated Imputation and Prediction (IIP) algorithm combining Correlation Dimension Estimation with SVR showed small average percentage prediction errors on three European ozone concentration time series, demonstrating particular strength in capturing nonlinear environmental dynamics [32].

## 2.3. Deep learning architectures

In a comparative evaluation on the Beijing Air Quality dataset, Hua et al. (2024) found that the Self-Attention-based Imputation for Time Series (SAITS) model consistently achieved lower MAE than RNN-based Bidirectional Recurrent Imputation for Time Series (BRITS) and other deep learning baselines across varying missing rates, confirming its effectiveness for time series imputation. The improvements were notable both in terms of accuracy and training efficiency compared to RNN-based methods [21].

Graph Neural Networks demonstrate particular strength for spatiotemporal environmental data. The Graph Recurrent Imputation Network (GRIN) achieves >20% MAE improvements over state-of-the-art methods by exploiting spatial correlations between monitoring stations through bidirectional message passing [33]. This approach proves especially valuable for sensor networks where geographical relationships contain crucial information for imputation.

LSTM-based imputation methods have demonstrated strong performance in recovering missing $PM_{2.5}$ data, particularly through architectural innovations. For example, the Transfer Learning-based LSTM Iterative Estimation (TLSTM-IE) model achieved 25–50% higher imputation accuracy compared to baseline methods on long-interval missing sequences in NYC $PM_{2.5}$ datasets by transferring knowledge from complete to incomplete sequences [34]. Separately, a standard LSTM-based imputation approach also showed substantial improvements over mean and moving average methods in $PM_{2.5}$ prediction tasks, reducing RMSE by up to 55–65% across multiple pollutants in Hefei, China [35]. These results confirm the robustness of LSTM architectures in handling both short and extended gaps in air quality time series.

The BRITS-ALSTM model by Wang et al. (2023) demonstrated $R^2 = 0.7454$ with 43.29% missing data rates at extreme locations on the Qinghai-Tibetan Plateau, representing a significant improvement over traditional methods in challenging environmental conditions [3]. Autoencoder-based approaches achieve up to 65% RMSE improvements over univariate methods for discontinuous and long-interval gaps [24,36].

Generative approaches offer unique advantages for uncertainty quantification. GAIN (Generative Adversarial Imputation Networks) and its variants provide not only point estimates but also uncertainty bounds, crucial for environmental decision-making [35,37]. The DEGAIN architecture, an enhanced version of Generative Adversarial Imputation Networks, introduces deconvolutional layers that improve reconstruction fidelity. Through adversarial training, DEGAIN achieves superior performance over baseline models, including autoencoders and MICE, particularly on multivariate air quality datasets. This reinforces the growing evidence that GAN-based approaches represent a leading strategy for restoring complex environmental time series [38].

## 2.4. Performance evaluation and method selection criteria

### 2.4.1. Missing data rate as primary selection criterion.
Empirical evidence from Seoul $PM_{2.5}$ datasets confirms that gap duration fundamentally determines optimal method selection, with simple approaches excelling for short intervals while sophisticated models become essential for extended periods [27]. For very short gaps (0.5–3 hours), classical methods like weighted k-nearest neighbors can outperform complex deep learning approaches [22]. For extreme missing rates exceeding 40%, Abiri et al. (2022) showed that spatiotemporal convolutional autoencoders achieve up to 65% RMSE improvement over univariate methods by exploiting correlations between monitoring stations.

Computational efficiency varies dramatically across method categories. Alkabbani et al. (2022) demonstrated that miss-Forest achieved 92.41% AQI prediction accuracy while maintaining reasonable computational costs compared to neural

networks [39]. The comprehensive Neural Computing and Applications study [40] revealed that Bi-LSTM with attention mechanisms processes Delhi pollution data in real-time while achieving superior accuracy to traditional LSTM. Real-time processing requirements favor computationally efficient architectures. Classical statistical methods demonstrate optimal balance between accuracy and computational cost for streaming applications [25], while advanced deep learning models like SAITS offer superior performance in batch processing scenarios [21]. Edge computing implementations support missing data rates up to 40% with acceptable latency for operational networks [41].

**2.4.2. Gap Pattern and Temporal Characteristics.** Wang et al. [3] definitively show that method selection must consider temporal characteristics: simple methods like forward fill excel for short gaps (≤6 hours), while sophisticated approaches like SARIMAX become necessary for longer periods (≥24 hours). Random Forest and XGBoost methods show consistent performance across gap durations, explaining their popularity in operational systems.

Geographic and environmental context affects performance stability. Studies across diverse locations (Kuwait [17], Tehran [28], Seoul [3], Beijing [21], European stations [26]) reveal that ensemble methods provide more robust performance across different climatic conditions and pollution patterns than single-algorithm approaches.

## 2.5. Specialized approaches for extreme data sparsity

Extremely sparse datasets demand fundamentally different strategies. Research by Chen et al. [42] on the FTLRI ("First Five & Last Three Logistic Regression Imputation") method shows that specialized temporal correlation models consistently achieve lowest MAE (4.46–5.68 range) at 40% missing rates, outperforming general-purpose algorithms. The FTLRI temporal correlation model specifically designed for high-missing-rate datasets demonstrates the value of purpose-built approaches.

Spatiotemporal methods prove crucial for extreme sparsity. The spatiotemporal convolutional autoencoder by Wardana et al. [36] achieves up to 65% RMSE improvement over univariate methods at 40% missing rates by exploiting correlations between nearby monitoring stations. This approach handles both short-interval and long-interval consecutive missing patterns effectively.

Tensor-based imputation techniques, such as Tucker decomposition, have demonstrated resilience under extreme data sparsity conditions – up to 90% missing – by leveraging multi-modal correlations. Tan et al. [43] report that their method successfully reconstructs entire missing days of traffic data, outperforming matrix-based approaches. These methods prove particularly valuable when entire days of environmental monitoring data are completely missing due to systematic equipment failures.

The BRITS-ALSTM architecture exhibits notable robustness in deployment scenarios characterized by extreme data sparsity. Wang et al. demonstrated that this approach maintains reasonable imputation performance ($R^2 = 0.7454$, MRE $= 0.2534$) even under the challenging conditions described earlier for high-altitude monitoring networks [3]. These results highlight the model's capacity to operate effectively where conventional techniques – such as mean filling, k-nearest neighbors, matrix factorization, and multiple imputation – experience significant degradation or operational failure. The bidirectional recurrent structure with attention mechanisms proves essential for extreme sparsity scenarios.

## 2.6. Implementation and deployment considerations

Real-time processing requirements favor specific architectural choices. Kalman filtering provides optimal balance for streaming applications, while Transformer models like SAITS offer batch processing advantages [25]. Edge computing implementations demonstrated by multiple studies support up to 40% missing data rates with acceptable latency for operational monitoring networks [44].

Ensemble-based machine learning approaches, particularly combinations of support vector regression (SVR) and Random Forest, have demonstrated operational robustness in scenarios with substantial data sparsity. Faramarzzadeh et

al. [41] report that such multi-model frameworks outperform traditional gap-filling techniques – including linear interpolation – when applied to daily precipitation datasets with missing rates ranging from 50% to 70%. However, these methods necessitate careful hyperparameter tuning and substantial computational resources to ensure stability and generalization across heterogeneous environmental conditions.

Model interpretability varies significantly across method categories. Classical statistical methods provide clear causal relationships, while deep learning approaches often function as black boxes. The N-BEATS inspired architecture addresses this limitation by explicitly decomposing time series into bias, slope, seasonality, and residual components with interpretable equations [45].

As model complexity increases, so does the demand for computational resources. Kim et al. [45] report that training deep neural networks for time-series imputation required a workstation equipped with 128 GB RAM and a single NVIDIA Titan X GPU (12 GB VRAM), with training times ranging from several hours depending on dataset size and missing rate. These resource constraints underscore the trade-off between model sophistication and inference efficiency – where simple architectures allow near real-time imputation, and deeper models necessitate batch processing strategies.

## 2.7. Summary

This comprehensive literature review reveals a mature field with clear performance hierarchies and practical guidance for method selection. Transformer-based approaches (particularly SAITS) represent the current state-of-the-art for most environmental monitoring applications, providing 12–38% improvements over previous methods while maintaining computational feasibility [21]. Classical statistical methods retain important niches for short-term gaps and resource-constrained deployments, while specialized high-sparsity methods become essential for extremely sparse datasets [2,46].

The field is moving toward ensemble approaches that combine multiple method strengths rather than seeking single optimal solutions. Future developments should focus on physics-informed methods that incorporate atmospheric dynamics, federated learning for privacy-preserving multi-network collaboration, and continual learning approaches that adapt to changing environmental conditions [37]. The convergence of high-performance computing and environmental monitoring creates unprecedented opportunities for accurate, real-time air quality data reconstruction across diverse deployment scenarios.

While significant progress has been achieved across all method categories, critical gaps remain for operational deployment in severely compromised monitoring networks. As identified in Section 1.4, existing approaches inadequately address extreme sparsity scenarios (<30% completeness) common in developing regions and harsh environments. This literature analysis confirms the need for specialized frameworks that can operate effectively under such challenging conditions.

## 3. Data and Methods

### 3.1. Data

**3.1.1. Data Source and Station Network.** This study utilized data from five air quality monitoring stations operated in Pavlodar, Kazakhstan, representing a comprehensive urban monitoring network covering different environmental and industrial zones. The monitoring stations were strategically positioned to capture spatial variability in air pollution across the city (Table 1).

A brief environmental characterization of the monitoring sites is provided to clarify their representativeness. The app_pspu station is located in the central urban area near educational and residential buildings with additional influence from a nearby railway corridor. The app_2pavlodar station represents a residential district situated closer to major industrial facilities, particularly the aluminum and energy complex. The app_center site is an urban-background location in the city center, characterized by high pedestrian activity, green corridors, and a dominant transport contribution. The app_metallurg

**Table 1. Pavlodar Air Quality Monitoring Network.**

| № | Station ID | Hosting Organization | Short Name | Address | Coordinates |
|---|---|---|---|---|---|
| 1 | 16101231 | Alkey Margulan University | app_pspu | 60 Olzhabay Batyr Street, Pavlodar | 52°17'56"N 76°57'18"E |
| 2 | 16101230 | Secondary School No. 12 named after K.N. Bekhozhin | app_2pavlodar | 60 Saltykov-Shchedrin Street, Pavlodar | 52°17'30"N 77°00'11"E |
| 3 | 16098828 | Secondary School No. 7 | app_center | 13 Victory Square, Pavlodar | 52°17'04"N 76°56'48"E |
| 4 | 16101232 | Secondary School No. 37 named after Zh. Tashenov | app_metallurg | 6/2 Vorushin Street, Pavlodar | 52°15'26"N 76°59'31"E |
| 5 | 16101233 | Secondary School named after M. Alimbayev | app_zaton | 17/2 Pavel Vasiliev Street, Pavlodar | 52°15'37"N 76°57'10"E |

station is positioned in a southern residential district located in closest proximity to metallurgical and energy facilities, making it the most industry-influenced site in the network. The app_zaton station represents a riverside residential microdistrict near the Irtysh floodplain with strong ventilation from the water–green corridor and industry located at moderate distances.

**3.1.2. Primary Target Variable.** The primary focus of this investigation was $PM_{2.5}$ concentration (μg/m³), representing fine particulate matter with aerodynamic diameter ≤2.5 micrometers. $PM_{2.5}$ serves as a critical indicator of air quality and poses significant health risks due to its ability to penetrate deep into respiratory systems and cross into the bloodstream [47]. All $PM_{2.5}$ measurements were collected using standardized monitoring equipment with automated quality control procedures.

**3.1.3. Temporal coverage and data resolution.** The study period spanned 14 months from May 23, 2024, to July 19, 2025, encompassing approximately 421 days of observations. This temporal coverage represents the complete operational period of the monitoring network from initial deployment to the time of manuscript preparation. The monitoring infrastructure was established specifically for this investigation, representing the first comprehensive air quality monitoring network deployed in Pavlodar. Prior to this study, no systematic large-scale air quality monitoring had been conducted in the city, making this dataset particularly valuable for understanding local pollution patterns. This temporal framework was designed to capture seasonal variations in air quality patterns and diverse meteorological conditions affecting particulate matter concentrations throughout multiple annual cycles.

Data collection involved continuous minute-level observations from monitoring equipment, which were subsequently aggregated into hourly averaged values for analysis.
Quality criteria for hourly aggregation were introduced:

- "Good quality": ≥40 valid minutes per hour

- "Weak quality": 30–39 valid minutes per hour;

- "Missing": <30 valid minutes per hour.

This aggregation approach ensured that hourly values represented statistically meaningful estimates while maintaining temporal resolution appropriate for atmospheric process analysis.

**3.1.4. Environmental and Meteorological Variables.** The monitoring stations were equipped with multi-parameter sensors AQS008A (Sichuan Weinasa Technology Co., Mianyang, China) [48] capable of simultaneous measurement of air quality and meteorological variables. Each station recorded the following parameters: air temperature, air humidity, $PM_{2.5}$, $PM_{10}$, $CO$, $SO_2$, $NO_2$, and $O_3$ concentrations. $PM_{2.5}$ measurements were obtained using optical sensors with the following specifications: measurement range 0–1000 μg/m³, resolution 1 μg/m³, and accuracy ±(10 μg/m³ + 10%) for concentrations below 500 μg/m³. These technical specifications ensure reliable quantification across the full range of atmospheric conditions encountered in urban environments, from background levels to severe pollution episodes. Database of initial measurements of $PM_{2.5}$ concentrations is available from the Zenodo database (https://doi.org/10.5281/zenodo.16937342).

Additional meteorological data were obtained from Pavlodar Airport meteorological station (UASP) [49] to supplement the on-site measurements. The complete meteorological dataset included:

 

- Air temperature (°C): Hourly ambient temperature measurements from station sensors

- Relative humidity (%RH): Hourly atmospheric moisture content from station sensors

- Wind speed (m/s): Hourly wind velocity measurements from airport meteorological station

- Wind direction (degrees): Hourly wind direction from airport station, converted from categorical descriptors to numerical values (0–360°)

All meteorological variables were resampled to hourly resolution using forward-fill methodology, followed by linear interpolation to address remaining gaps. The synchronized meteorological data were merged with air quality measurements using temporal alignment procedures, ensuring consistency across all variables.

### 3.1.5. Computational Methods and Software Implementation.

All calculations and analyses in this study were performed programmatically using Python 3.8+ as the primary programming language. The computational framework leverages several key scientific computing libraries, including pandas (version 2.0+) for time series manipulation and data processing, numpy (version 1.24+) for numerical computations, scikit-learn (version 1.3+) for machine learning algorithms and evaluation metrics, xgboost (version 2.0+) for the gradient boosting implementation of the model, matplotlib (version 3.7+) and seaborn (version 0.12+) for comprehensive data visualization, and openpyxl (version 3.1+) for Excel file handling.

The software architecture follows a modular design principle, with code organized into independent scripts and reusable modules to ensure reproducibility and maintainability. All code underlying the findings in this study is publicly available at: https://github.com/ruslan-saf/air-quality-imputation.

## 3.2. Data preprocessing

The data preprocessing pipeline consists of several sequential stages designed to transform raw sensor measurements into analysis-ready datasets (Fig 1). The workflow incorporates quality control measures at multiple levels, from minute-level duplicate removal to hourly-level outlier detection, ensuring data integrity while preserving temporal patterns essential for imputation model training.

### 3.2.1. Outliers detection.

A comprehensive two-stage outlier detection system was implemented to identify and handle anomalous values in $PM_{2.5}$ measurements while preserving legitimate pollution episodes.

**Stage 1: Minute-level preprocessing.** The first stage addressed fundamental data quality issues through systematic identification of anomalous values. Duplicate measurements sharing identical timestamps and station identifiers were removed to ensure data integrity. Physically impossible values were identified using absolute bounds criteria, where $PM_{2.5}$ concentrations below 0 µg/m³ or exceeding 1000 µg/m³ were flagged for removal. These extreme values typically indicate sensor malfunctions or calibration errors.

**Stage 2: Hourly-level filtering.** Secondary filtering employed a hierarchical series of three complementary filters designed to address different types of measurement anomalies

Filter 1: Station-specific upper limits

Station-specific thresholds were calculated using the formula (1):

$$thr_i = min \left( 10^{Q_{0.995}[log_{10}(x_i+1)]} - 1, \ 900 \right) \ \mu g \ m^{-3}$$

(1)

where $Q_{0.995}$ represents the 99.5th percentile calculated after log-transforming x+1 to dampen heavy tails. This approach accounts for the highly skewed distribution of $PM_{2.5}$ concentrations while establishing station-specific upper bounds that reflect local pollution characteristics. Value of 900 µg m⁻³ serves as sensor-saturation filter. This threshold ensures removal of measurements beyond reliable instrument range.

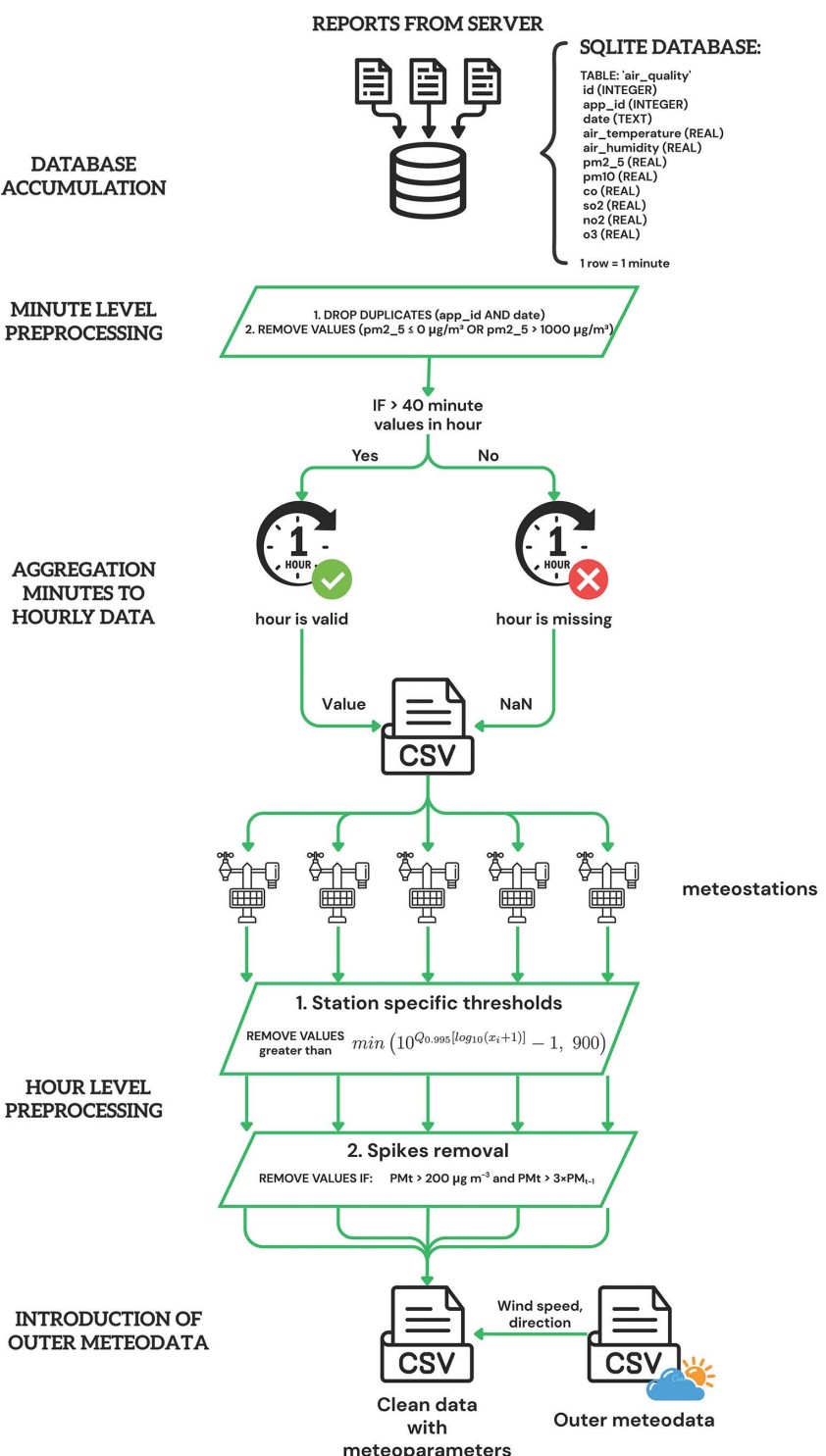

**REPORTS FROM SERVER**

**SQLITE DATABASE:**

TABLE: 'air_quality'
id (INTEGER)
app_id (INTEGER)
date (TEXT)
air_temperature (REAL)
air_humidity (REAL)
pm2_5 (REAL)
pm10 (REAL)
co (REAL)
so2 (REAL)
no2 (REAL)
o3 (REAL)

1 row = 1 minute

**DATABASE ACCUMULATION**

**MINUTE LEVEL PREPROCESSING**

1. DROP DUPLICATES (app_id AND date)
2. REMOVE VALUES (pm2_5 ≤ 0 µg/m³ OR pm2_5 > 1000 µg/m³)

IF > 40 minute values in hour

Yes    No

**AGGREGATION MINUTES TO HOURLY DATA**

hour is valid    hour is missing

Value    NaN

CSV

meteostations

**1. Station specific thresholds**

REMOVE VALUES greater than $min\left(10^{Q_{0.995}[log_{10}(x_i+1)]} - 1,\ 900\right)$

**HOUR LEVEL PREPROCESSING**

**2. Spikes removal**

REMOVE VALUES IF:    PMt > 200 µg m⁻³ and PMt > 3×PMt₋₁

**INTRODUCTION OF OUTER METEODATA**

Wind speed, direction

CSV — Clean data with meteoparameters

CSV — Outer meteodata

**Fig 1. Data preprocessing workflow for PM$_{2.5}$ monitoring network.**

Filter 2: Single-step spikes

Isolated measurement spikes were identified and removed using dual criteria (2):

$$PM_t > 200\ \mu g\ m^{-3}\ and\ PM_t > 3 \times PM_{t-1} \qquad (2)$$

This filter captures isolated jumps caused by dust bursts or electronic noise that do not persist in subsequent measurements, effectively distinguishing between measurement artifacts and genuine pollution episodes.

The hierarchical filtering approach systematically addressed different sources of measurement error while preserving legitimate atmospheric pollution events. Detected outliers were replaced with NaN values to maintain temporal structure while enabling subsequent imputation procedures (Table 2).

**3.2.2. Data Quality Characteristics.** Following outlier detection and removal, the monitoring network exhibited substantial heterogeneity in data completeness, reflecting real-world challenges in environmental monitoring systems. Data completeness varied significantly across stations (Table 3).

The monitoring network demonstrated considerable variability in data quality, with mean data completeness of 46.3% across all stations. The complete dataset comprised approximately 50,667 hourly observations over the 15-month period (May 23, 2024 to July 19, 2025), with 23,471 valid measurements and 27,196 missing observations. The network exhibited significant spatial heterogeneity in $PM_{2.5}$ concentrations, with station-specific mean values ranging from 17.5 µg/m³ (app_zaton) to 84.8 µg/m³ (app_metallurg).

Data completeness patterns revealed two distinct station categories. Higher-performing stations (app_center, app_2pavlodar, and app_pspu) achieved 56–57% completeness. Lower-performing stations (app_metallurg and app_zaton) exhibited 23–37% completeness throughout the entire monitoring period.

**Table 2. Outlier detection and removal results by monitoring station.**

| Station | Valid Observations | 99.5th Percentile (log₁₀) | Station Threshold (µg/m³) | Station Threshold Removals | Spike Removals | Total Removed | Removal Rate (%) |
|---|---|---|---|---|---|---|---|
| app_center | 5 677 | 2.2049 | 159.3 | 29 | 2 | 31 | 0.55 |
| app_2pavlodar | 5 869 | 2.4498 | 280.7 | 30 | 2 | 32 | 0.55 |
| app_pspu | 5 960 | 2.9666 | 900.0 | 37 | 98 | 135 | 2.27 |
| app_metallurg | 2 500 | 2.9759 | 900.0 | 99 | 43 | 142 | 5.68 |
| app_zaton | 3 811 | 2.4184 | 261.0 | 20 | 3 | 23 | 0.60 |
| **Total** | **23 817** | – | – | **215** | **148** | **363** | **1.52** |

**Note:** The 99.5th percentile values are calculated in $\log_{10}(PM_{2.5} + 1)$ space and then back-transformed to µg/m³ according to formula (1). Station thresholds were calculated using formula (1). Spike removals identify isolated measurements exceeding 200 µg/m³ and three times the preceding hourly value by criteria (2).

**Table 3. Data quality characteristics and $PM_{2.5}$ concentration statistics for the five-station monitoring network in Pavlodar, Kazakhstan[1].**

| Station | Data Completeness[2] (%) | Period (days) | Mean $PM_{2.5}$ (µg/m³) | Max $PM_{2.5}$ (µg/m³) |
|---|---|---|---|---|
| app_center | 56.0 | 420 | 25.1 | 159.2 |
| app_2pavlodar | 57.5 | 422 | 33.8 | 279.4 |
| app_pspu | 57.4 | 422 | 64.3 | 880.6 |
| app_metallurg | 23.3 | 421 | 84.8 | 896.8 |
| app_zaton | 37.4 | 422 | 17.5 | 255.5 |

[1]All statistics calculated after removal of statistical outliers.

[2]Data completeness represents the percentage of valid hourly measurements over the whole study period.

The observed data completeness patterns reflect typical challenges in environmental monitoring networks, including equipment malfunctions, calibration periods, power outages, and communication failures [1,4]. Notably, station app_metallurg exhibited extremely low completeness (23.3%) representing the type of severely compromised monitoring infrastructure commonly encountered in operational networks. Despite these challenges, the dataset provides a realistic testbed for evaluating imputation methods under varying data quality scenarios, from moderately incomplete to severely sparse monitoring conditions.

**3.2.3. Data compression strategies.** Due to the extreme sparsity of the monitoring data, model training was conducted on time series after removing missing values (NaN), effectively compressing the temporal sequences into dense arrays of valid observations suitable for machine learning algorithms. Synthetic gaps for training, testing, and benchmarking imputation methods were subsequently created within these dense datasets to simulate realistic missing data scenarios while maintaining ground truth values for performance evaluation.

Data compression in the research represents the process of removing rows with NaN from $PM_{2.5}$ time series to create dense training datasets. This technical operation transitions sparse time series with gaps to continuous arrays of valid observations. For example, if an original time series contains 10,000 hourly measurements with 4,000 missing values, compression yields an array of 6,000 consecutive valid observations without temporal structure.

Two distinct compression approaches were evaluated to assess the impact of data preprocessing strategies on imputation performance.

Selective compression uses a quality-first approach with strict filtering criteria applied before NaN removal. The method evaluates monthly data completeness against a 70% threshold and requires 24 consecutive valid hours. Only months meeting both criteria are retained, ensuring data quality while reducing dataset size. Stations must have at least three months above the 70% threshold to be included in the analysis.

Full compression adopts a quantity-first approach, directly removing all NaN values without preliminary quality filtering. This method preserves all available valid measurements regardless of their temporal context or surrounding data quality, maximizing the utilization of sparse monitoring data (Table 4).

The results demonstrate fundamental differences between approaches. Selective compression achieved high completeness rates (>90%) in retained datasets but resulted in substantial data loss, with stations retaining only 21–45% of original observations. Most critically, app_metallurg was completely excluded due to insufficient monthly completeness. Full compression maintained perfect completeness (100%) while preserving all originally valid measurements, demonstrating universal applicability across all monitoring stations regardless of data quality characteristics.

The results demonstrate fundamental differences between approaches. Selective compression achieved completeness rates of 82.5–87.8% in retained datasets but led to significant monthly data exclusion. Stations with poor initial data quality experienced the greatest losses: app_zaton retained only 4 out of 15 months, while app_metallurg was completely excluded from selective compression due to failing quality criteria. Even stations with moderate completeness lost substantial temporal coverage, with app_2pavlodar retaining 6 months and app_pspu retaining 7 months. Only app_center, with the highest initial completeness, preserved 8 months. In contrast, full compression maintained perfect completeness

**Table 4. Impact of compression strategies on data completeness.**

| Station | Original completeness (%) | After selective compression (%) | After full compression (%) | Valid Months |
|---|---|---|---|---|
| app_center | 56.0 | 85.1 | 100.0 | 8/15 |
| app_2pavlodar | 57.5 | 82.5 | 100.0 | 6/15 |
| app_pspu | 57.4 | 85.0 | 100.0 | 7/15 |
| app_metallurg | 23.3 | 0.0 | 100.0 | 0/15 |
| app_zaton | 37.4 | 87.8 | 100.0 | 4/15 |

(100%) while preserving all originally valid measurements, demonstrating universal applicability across all monitoring stations regardless of data quality characteristics.

Thus, selective compression aims to ensure high-quality, temporally coherent training data by retaining only months that meet predefined completeness and continuity requirements. In contrast, full compression aims to maximize data availability by preserving all valid observations regardless of temporal structure.

For clarity, the decision rules for both compression strategies are summarized below.

### 3.2.4. Algorithms of compression.

Selective Compression:

1. For each month in the time series:

2. Compute monthly completeness.

3. If completeness ≥ 70%:

4. Check whether a 24-hour valid sequence exists.

5. If yes, mark this month as eligible.

6. Retain only eligible months.

7. Remove all NaN values within retained months.

Full Compression:

1. Remove all rows containing NaN values from the time series.

2. Return the resulting dense sequence of valid observations.

### 3.2.5. Feature Scaling for ML Models.

Feature scaling was implemented using the StandardScaler algorithm from scikit-learn to ensure optimal performance of machine learning algorithms. The standardization process applied the z-score transformation formula (3):

$$z = \frac{x - \mu}{\sigma}$$

(3)

where $\mu$ represents the mean and $\sigma$ the standard deviation of each feature. This transformation was applied to all numerical variables including $PM_{2.5}$ concentrations, meteorological parameters, and temporal features.

Standardization parameters (means and standard deviations) were estimated from the complete set of available non-missing observations in the dense time series prior to synthetic gap generation. Additional sensitivity tests with normalization fitted solely on the training subset yielded very similar performance, so the simpler global standardization scheme was retained without affecting the substantive conclusions.

The standardization procedure was essential for addressing the heterogeneous scales present in environmental monitoring data, where $PM_{2.5}$ concentrations (measured in µg/m³) exhibit substantially different ranges compared to meteorological variables such as relative humidity (percentage) and wind speed (m/s). The implementation ensured that no single variable would dominate the learning process due to scale differences, thereby improving algorithm convergence and prediction stability.

## 3.3. Benchmarking Methodology and Synthetic Gap Generation

The benchmarking methodology follows a systematic workflow to ensure fair comparison between developed Dynamic Sequence-to-Sequence model implemented using eXtreme Gradient Boosting (DynamicSeq2SeqXGB) and classical imputation methods (Fig 2). The process encompasses two parallel compression strategies, unified model training,

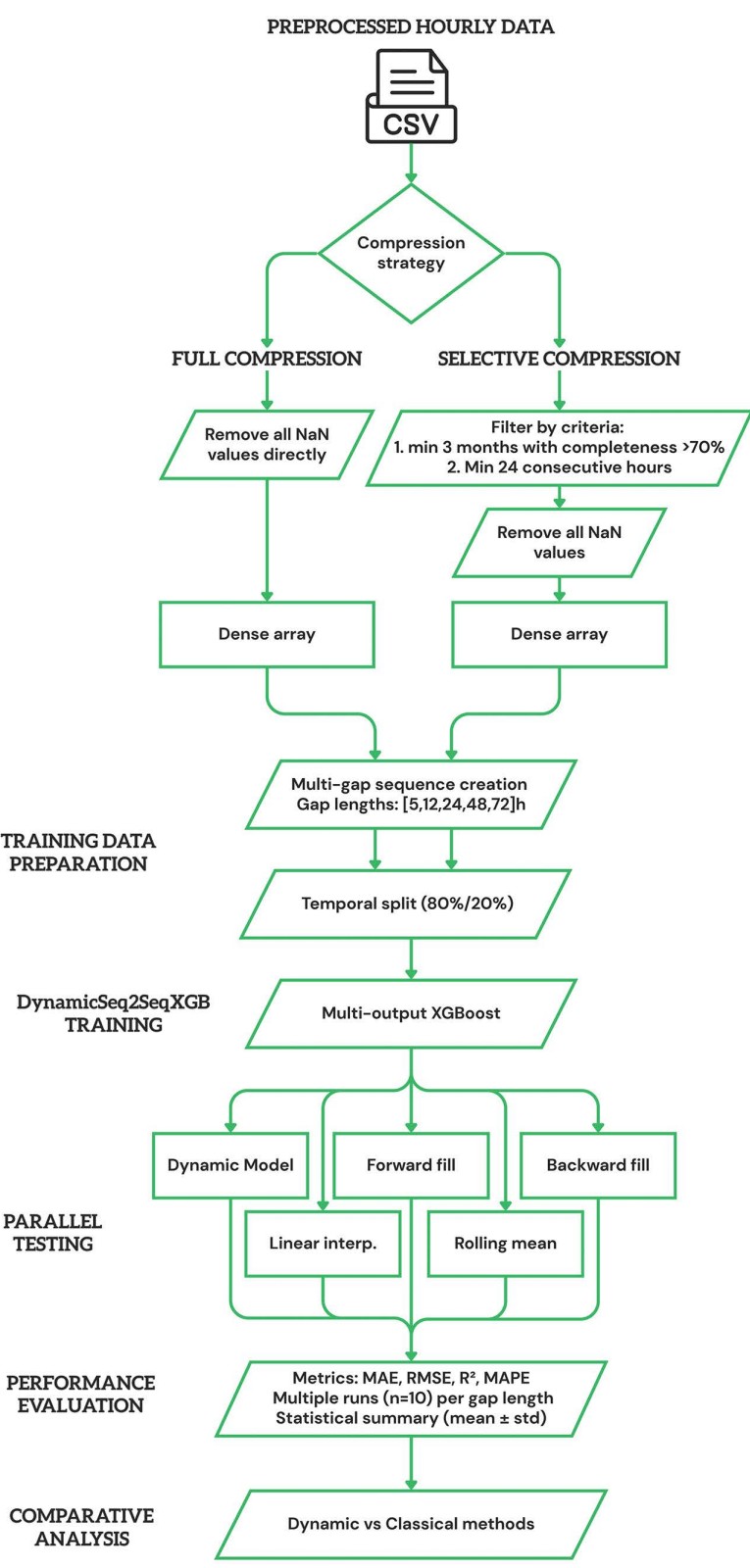

**Fig 2. Comprehensive benchmarking methodology workflow for imputation method comparison.**

synchronized testing on identical synthetic gaps, and comprehensive statistical evaluation across multiple experimental runs.

**3.3.1. Benchmarking Framework Overview.** The benchmarking methodology ensures fair comparison between DynamicSeq2SeqXGB and classical imputation methods through controlled synthetic gap testing. The framework operates on compressed dense data arrays, creating identical missing data scenarios for all methods to eliminate evaluation bias. This approach enables systematic assessment of imputation quality across varying gap characteristics while maintaining ground truth values for accurate performance measurement.

**3.3.2. Synthetic Gap Generation Protocol.** Synthetic gaps are systematically generated within dense data portions to simulate realistic monitoring system failures. The algorithm creates gaps of predetermined lengths (5, 12, 24, 48, 72 hours) representing different failure scenarios: sensor malfunctions, calibration periods, power outages, communication failures, and extended equipment downtime.

Gap positions are randomly selected with constraints ensuring adequate contextual information. Each gap requires 32 hours of valid data before and after the missing period, establishing a minimum sequence length of 136 hours for 72-hour gaps. This context requirement ensures sufficient temporal information for sequence-to-sequence reconstruction while preventing boundary effects.

The synthetic gap generation follows specific parameters designed to reflect operational monitoring conditions:

1. Gap frequency: 5% of total data points, calculated as (data_length × 0.05)/ gap_length

2. Gap lengths: 5, 12, 24, 48, 72 hours covering short-term to extended outages

3. Context preservation: 32 hours minimum before and after each gap

4. Non-overlapping constraint: Gaps maintain minimum separation to prevent interference

5. Reproducibility: Fixed random seed sequence (1–10) across ten experimental cycles ensures reproducible gap patterns while preserving inter-run variability.

Each generated gap undergoes validation to ensure experimental integrity. Gaps failing validation criteria – insufficient context boundaries, overlap with existing gaps, or proximity to data boundaries – are excluded, and alternative positions are selected to maintain target gap frequency.

**3.3.3. Multi-Gap Training Data Preparation.** The model training incorporates all gap lengths simultaneously to develop adaptive reconstruction capabilities. Training sequences are extracted using sliding window methodology across the dense data array, creating sequence-to-sequence pairs for each gap length configuration.

For each potential gap position, the algorithm extracts: (1) pre-gap context window (32 hours), (2) target sequence (gap content), (3) post-gap context window (32 hours), (4) metadata features (gap length, dynamic context size, position indicators). This multi-gap approach enables the model to learn length-specific reconstruction strategies within a unified framework.

Training and validation sets are created using temporal splitting (80%/20%) to preserve chronological order and prevent data leakage. The split maintains temporal sequence integrity, ensuring that validation data represents future time periods relative to training data. This approach reflects realistic deployment scenarios where models must predict future missing values based on historical patterns.

To fully eliminate any risk of temporal leakage, all sliding-window training sequences were constructed strictly within the 80% training segment, and no input window was allowed to cross the train–validation boundary. Validation sequences were generated exclusively from future timestamps after the split. Synthetic gaps were inserted independently within each segment only after temporal separation, ensuring that neither direct values nor contextual information from the validation horizon influenced the training process.

### 3.3.4. Synchronized Evaluation Framework.

All imputation methods are tested on identical synthetic gaps to ensure fair comparison. The testing framework creates gap masks within the dense data array and applies each method to the same missing positions. This synchronized approach eliminates variations in gap characteristics that could bias comparative results.

The evaluation process involves: (1) gap mask application to dense data, (2) DynamicSeq2SeqXGB prediction for masked positions, (3) classical method application to identical gaps, (4) performance metric calculation for all methods, (5) statistical comparison across multiple runs.

Four classical imputation methods serve as baseline comparisons: Forward Fill, Backward Fill, Linear Interpolation, and Rolling Mean (24-hour window). Detailed descriptions of these methods are provided in Section 3.4. All baseline methods are applied to identical synthetic gaps to ensure fair comparison, with consistent preprocessing and evaluation protocols across all methods.

### 3.3.5. Performance Evaluation and Statistical Analysis.

Model performance is assessed using four complementary metrics capturing different aspects of reconstruction quality:

Mean Absolute Error (MAE) measures average absolute deviation between predicted and actual values, providing interpretable error magnitude in original units ($\mu g/m^3$). This metric gives equal weight to all prediction errors regardless of magnitude.

Root Mean Square Error (RMSE) emphasizes larger prediction errors through squaring, making it sensitive to outliers and extreme values. RMSE values consistently exceed MAE values, with larger differences indicating greater prediction variability.

Coefficient of Determination ($R^2$) quantifies the proportion of variance explained by predictions, ranging from 0 to 1. Higher $R^2$ values indicate better capture of temporal patterns and overall reconstruction quality.

Mean Absolute Percentage Error (MAPE) provides scale-independent error assessment by normalizing absolute errors relative to actual values. This metric facilitates comparison across different pollutant concentration ranges.

MAE, RMSE, $R^2$, and MAPE were selected because they are standard metrics for evaluating gap-filling accuracy in environmental time-series reconstruction and allow comparison with prior studies. MAE and RMSE quantify absolute and squared errors, respectively, offering complementary views of prediction deviations. $R^2$ captures the proportion of variance explained by the model. MAPE is included for interpretability in percentage terms, but its known limitation – instability when true values are low – is acknowledged, and therefore MAPE is interpreted with caution in periods of very low $PM_{2.5}$ concentrations.

Muliple experimental runs (n = 10) are conducted for each gap length to assess statistical significance and measurement reliability. Each run employs different random seeds for gap generation, creating independent synthetic gap sets while maintaining identical gap frequency and length distributions.

Results are summarized using descriptive statistics (mean and standard deviation) across runs. This multi-run approach ensures that observed performance differences reflect systematic patterns rather than random variation, providing confidence intervals for performance estimates across different experimental conditions.

The benchmarking protocol is applied consistently across all monitoring stations, enabling assessment of method performance under varying data quality conditions. Stations with different completeness rates (23.3% to 57.5%) provide natural experiments for evaluating method robustness under operational constraints.

This cross-station approach reveals method sensitivity to data sparsity and helps identify optimal application domains for different imputation strategies. Results are aggregated across stations to provide system-level performance assessments relevant for operational monitoring network deployment.

This comprehensive benchmarking methodology provides robust comparative assessment of imputation methods while maintaining scientific rigor and operational relevance for environmental monitoring applications.

## 3.4. Classical imputation methods

To establish performance benchmarks and validate the effectiveness of DynamicSeq2SeqXGB, four widely-used classical imputation methods commonly applied in environmental monitoring systems were implemented. These methods represent the current standard practice in operational air quality networks due to their computational simplicity and ease of implementation.

**3.4.1. Linear Interpolation.** Linear interpolation estimates missing values by creating straight-line connections between adjacent known data points. The interpolated value is calculated as:

$$PM_{2.5}(t) = PM_{2.5}(t_1) + \left(\frac{t - t_1}{t_2 - t_1}\right) \cdot (PM_{2.5}(t_2) - PM_{2.5}(t_1))$$

$$(4)$$

where $t_1$ and $t_2$ are the nearest available timestamps before and after the gap. This method assumes linear temporal trends and performs well for short gaps but degrades significantly with increasing gap length, as it cannot capture non-linear patterns or diurnal variations characteristic of atmospheric pollutant concentrations [50].

**3.4.2. Rolling Mean.** The rolling mean method fills missing values using the average of a sliding window of available observations. A 24-hour window (12 hours before and after each missing point) was implemented to capture short-term temporal patterns while maintaining sensitivity to recent changes:

$$PM_{2.5}(t) = \frac{1}{n} \sum_{i=1}^{n} PM_{2.5}(t_i)$$

$$(5)$$

where $n$ represents the number of available observations within the 24-hour window, and $t_i$ are the timestamps of valid measurements [51]. While the 24-hour window effectively captures temporal patterns and provides stable estimates, this averaging approach may underestimate extreme pollution events.

**3.4.3. Forward Fill.** Forward fill (also known as "last observation carried forward" or LOCF) propagates the most recent valid observation to fill subsequent missing values:

$$PM_{2.5}(t) = PM_{2.5}(t_{last})$$

$$(6)$$

where $t_{last}$ is the timestamp of the last available measurement before the gap. This method assumes temporal persistence of atmospheric conditions and performs reasonably well for short gaps but becomes increasingly unrealistic for extended periods, as it cannot account for natural temporal variations in pollutant concentrations [52]. LOCF is widely implemented in time-series databases and environmental monitoring systems as a computationally efficient gap-filling technique, particularly useful when data shows relatively stable trends.

**3.4.4. Backward Fill.** Backward fill propagates the next available observation backward to fill preceding missing values:

$$PM_{2.5}(t) = PM_{2.5}(t_{next})$$

$$(7)$$

where $t_{next}$ is the timestamp of the first available measurement after the gap. This method assumes that future conditions can inform past missing values, which may be appropriate when data processing occurs retrospectively. However, it violates causality principles and may introduce artificial patterns when applied to real-time monitoring scenarios [52]. Studies have shown that backward fill can effectively handle downward trends in environmental time series, though it may introduce artificial jumps or dips in the data.

**3.4.5. Implementation Details.** All classical methods were implemented using pandas library functions with consistent handling of boundary conditions and edge cases. For gaps occurring at the beginning or end of time series, forward fill and backward fill were applied as fallback methods for linear interpolation and rolling mean, respectively. The methods were applied independently to each station's $PM_{2.5}$ time series, maintaining the temporal structure of the original datasets.

These baseline methods serve as performance benchmarks for evaluating the effectiveness of DynamicSeq2SeqXGB. The comparison enables quantification of the added value provided by machine learning approaches over conventional gap-filling techniques currently employed in operational environmental monitoring systems.

## 3.5 DynamicSeq2SeqXGB Model Architecture

The complete processing workflow of DynamicSeq2SeqXGB is illustrated in Fig 3, which demonstrates the sequence-to-sequence transformation from sparse environmental time series to gap-filled $PM_{2.5}$ data through adaptive context processing and multi-output prediction.

**3.5.1. Core Architecture: XGBoost MultiOutputRegressor Foundation.** DynamicSeq2SeqXGB employs XGBoost as its fundamental regression engine, implemented through scikit-learn's MultiOutputRegressor wrapper to handle variable-length gap prediction. The model uses XGBRegressor with 50 estimators, maximum depth of 6, learning rate of 0.1, and squared error objective function, optimized for environmental time series reconstruction. No grid search or randomized hyperparameter optimization was performed. These values were selected based on prior exploratory experiments conducted during model prototyping and were chosen to balance predictive performance with computational efficiency, which is essential for operational deployment in resource-constrained monitoring systems. This fixed-parameter configuration also ensures reproducibility and isolates the effects of data sparsity and sequence length from confounding hyperparameter variance.

The gradient boosting framework constructs decision trees sequentially, where each tree corrects residual errors from the previous ensemble. The relatively conservative estimator count (50) prevents overfitting on sparse environmental datasets while maintaining sufficient model complexity to capture non-linear relationships between meteorological variables and pollutant concentrations. The MultiOutputRegressor architecture internally creates separate XGBRegressor instances for each output position in the sequence – for a 4-hour gap, it generates four regressors (XGBRegressor_1 through XGBRegressor_4) that all use the same input features but are trained to predict different temporal positions within the gap.

**3.5.2. Sequence-to-Sequence Architecture Implementation.** The sequence-to-sequence paradigm is implemented through structured feature engineering that creates encoder-decoder functionality using tabular data rather than traditional neural architectures. The "encoder" component extracts contextual information from available data surrounding each gap, while the "decoder" generates sequential predictions for missing time steps.

The model employs adaptive context windows with maximum capacity C_max = 32 time steps. For each gap, dynamic context length C_dynamic is calculated as min(gap_length × 3, 32) for gaps ≤ 10 hours, or fixed at 32 for longer gaps. This adaptive sizing ensures optimal information utilization: short gaps receive focused local context, while extended gaps rely on broader temporal patterns.

For a 4-hour gap, C_dynamic = 4 × 3 = 12 hours, meaning only 12 hours of actual data are extracted from each side of the gap (as demonstrated in Fig 3). Context extraction processes both pre-gap and post-gap sequences, where pre-gap context captures recent temporal trends and immediate environmental conditions, while post-gap context provides boundary conditions for reconstruction.

Both contexts are padded to C_max dimensions using zero-padding to maintain consistent input dimensionality across different gap characteristics. The padding strategy differs for left and right contexts:

• Left context: Zero-padding is applied at the beginning, resulting in [20 zero hours] + [12 real data hours] for a 4-hour gap

• Right context: Zero-padding is applied at the end, resulting in [12 real data hours] + [20 zero hours]

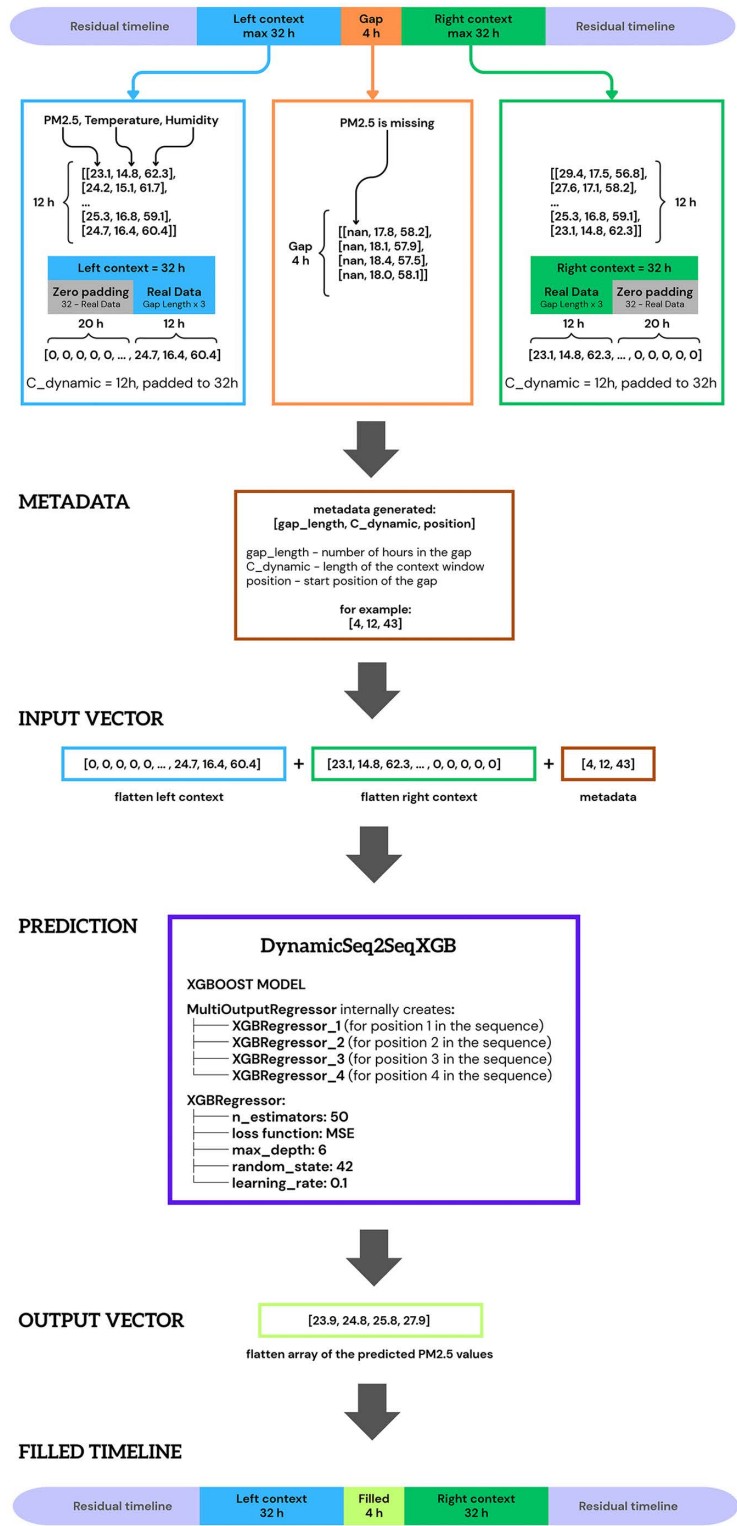

**SPARSE TIMELINE**

**METADATA**

**INPUT VECTOR**

**PREDICTION**

**OUTPUT VECTOR**

**FILLED TIMELINE**

**Fig 3. DynamicSeq2SeqXGB sequence processing workflow.**

This asymmetric padding preserves the temporal relationship between contexts and the gap, ensuring that real data remains adjacent to the missing period.

Sequential dependencies in DynamicSeq2SeqXGB are handled through sliding context windows and simultaneous multi-step prediction rather than autoregressive recursion. For each artificial gap, fixed-length left and right context segments are extracted from the dense time series, transformed into a single feature vector and mapped directly to the full sequence of missing $PM_{2.5}$ values in that gap. Within the MultiOutputRegressor framework, each XGBRegressor instance learns to predict one specific position inside the gap, so the model generates the entire imputed sequence in a single forward pass conditioned on the surrounding context.

Training targets consist only of originally observed $PM_{2.5}$ values that are temporarily masked to form artificial gaps, as described in the benchmarking protocol. For each output position, the model minimizes the standard squared-error objective between predicted and true values, and no loss is computed outside the masked interval. Consequently, masked outputs do not require a special treatment in the loss function: the learning problem is formulated as supervised regression on the removed observations within each gap.

**3.5.3. Feature Engineering Pipeline.** The model constructs comprehensive feature vectors through systematic combination of temporal, environmental, and metadata components (Fig 3). Each time step contains multiple environmental parameters, creating matrices of dimensions 32 × n_features for each context. While Fig 3 illustrates the process using three parameters ($PM_{2.5}$, temperature, humidity) for clarity, the actual implementation incorporates a broader set of features: $PM_{10}$, $SO_2$, $NO_2$, $CO$, $O_3$ concentrations, meteorological variables (air temperature, air humidity, wind speed, wind direction), and temporal features (hour of day, season indicators).

The final input vector is constructed through concatenation of flattened contexts and metadata, as illustrated in Fig 3:

1. Left context flattening: 32 × n_features matrix → flattened vector

2. Right context flattening: 32 × n_features matrix → flattened vector

3. Metadata vector: 3-element array containing [gap_length, C_dynamic, position], where:

• gap_length: number of hours in the missing sequence;

• C_dynamic: actual context window size used (before padding);

• position: absolute position of gap start in the original time series.

4. Final concatenation: For the example shown in Fig 3 with 3 features ($PM_{2.5}$, temperature, humidity), the total input vector size = 96 + 96 + 3 = 195 features. In practice, with the full feature set listed above, the vector size scales proportionally with the number of environmental and temporal parameters. Thus, the final input vector is a flatten array [left_flat, right_flat, metadata].

**3.5.4. Dynamic Gap-Length Adaptation Mechanisms.** The dynamic component manifests through gap-aware training and prediction strategies that automatically adjust model behavior based on missing data characteristics. Unlike traditional approaches that apply uniform strategies regardless of gap properties, DynamicSeq2SeqXGB implements distinct processing pathways optimized for different temporal scales.

The model trains on artificial gaps created from dense (non-missing) portions of the dataset. Multiple gap lengths (5, 12, 24, 48, 72 hours) are simultaneously incorporated during training, exposing the model to diverse reconstruction scenarios. This multi-gap training strategy ensures robust performance across the full spectrum of missing data scenarios encountered in operational monitoring.

During prediction, gap length detection triggers automatic pathway selection: Short Gaps (≤ 6 hours): C_dynamic ranges from 15–18 hours (5 × 3–6 × 3), emphasizing high-frequency temporal features and immediate environmental correlations through focused local context.

Medium Gaps (6–10 hours): C_dynamic ranges from 18–30 hours, focusing on diurnal patterns and meteorological drivers as rolling statistics and daily cyclical features become primary predictors.

Extended Gaps (> 10 hours): Context windows are capped at C_dynamic = 32 hours for both pre-gap and post-gap segments (64 hours total context). The model's maximum single-inference capacity is 72 hours of gap prediction. For gaps exceeding this limit, a segmented approach divides the gap into sequential 72-hour chunks. Each chunk receives independent context windows from the original time series, and predictions are concatenated to reconstruct the complete sequence.

A single DynamicSeq2SeqXGB model is therefore fitted on a combined training set that includes examples from all predefined gap lengths, with gap_length and C_dynamic provided as metadata features, enabling the model to adapt its predictions to varying outage durations.

A comprehensive benchmarking of 46 gap-filling models, including Random Forest, missForest-type algorithms, GRU/LSTM networks, autoencoders, and hybrid architectures, was performed in our previous study [53]. That analysis demonstrated the superior accuracy of tree-based bidirectional Seq2Seq models and the practical advantages of dynamic multivariate frameworks. Therefore, the present work focuses on evaluating data compression strategies and station-specific behaviour, while adopting DynamicSeq2SeqXGB as a previously validated state-of-the-art model rather than re-running the entire suite of ML baselines.

## 4. Results

### 4.1. Data Quality and Completeness Assessment of the Pavlodar Monitoring Network

Comprehensive assessment of data quality across the five-station monitoring network revealed substantial heterogeneity in measurement completeness and gap patterns throughout the 15-month study period. The analysis identified significant variations in data availability both temporally and spatially, with important implications for imputation strategy selection. Fig 4 presents comprehensive gap pattern analysis across both temporal and statistical dimensions, providing critical insights into the operational challenges faced by each monitoring station.

Data completeness patterns varied considerably across monitoring stations, ranging from 23.3% (app_metallurg) to 57.5% (app_2pavlodar), with a network-wide mean of 46.3%. The temporal distribution of missing data exhibited distinct seasonal patterns, with winter months (December-February) showing decreased completeness rates across most stations, likely attributable to equipment malfunctions during severe weather conditions and increased maintenance requirements during high-pollution episodes.

Gap duration analysis revealed highly skewed distributions with substantial heterogeneity across monitoring stations. Mean gap lengths varied from 3.5 hours (app_pspu) to 32.1 hours (app_metallurg), with considerable variation in gap frequency from 127 gaps (app_center) to 370 gaps (app_pspu). Maximum gap durations demonstrated extreme variability, ranging from 307 hours at app_metallurg to 3,268 hours (136 days) at app_zaton. The gap pattern analysis indicates that while most stations experienced frequent short interruptions, extended outages substantially impact overall data availability and present significant challenges for conventional interpolation methods.

Station-specific characteristics demonstrated clear clustering into distinct operational categories. Higher-performing stations (app_center, app_2pavlodar, app_pspu) achieved 56–58% completeness with consistent monitoring patterns, while lower-performing stations (app_metallurg, app_zaton) exhibited 23–37% completeness with more erratic data collection. Station app_metallurg showed the most challenging operational profile with only 23.3% completeness despite having fewer total gaps (242), indicating prolonged equipment failures.

Two data compression strategies produced markedly different outcomes. Selective compression achieved 82.5–87.8% completeness in retained datasets while dramatically reducing data volume, with compression ratios ranging from 0.239 (app_zaton) to 0.524 (app_center). Most critically, app_metallurg was completely excluded due to insufficient monthly data

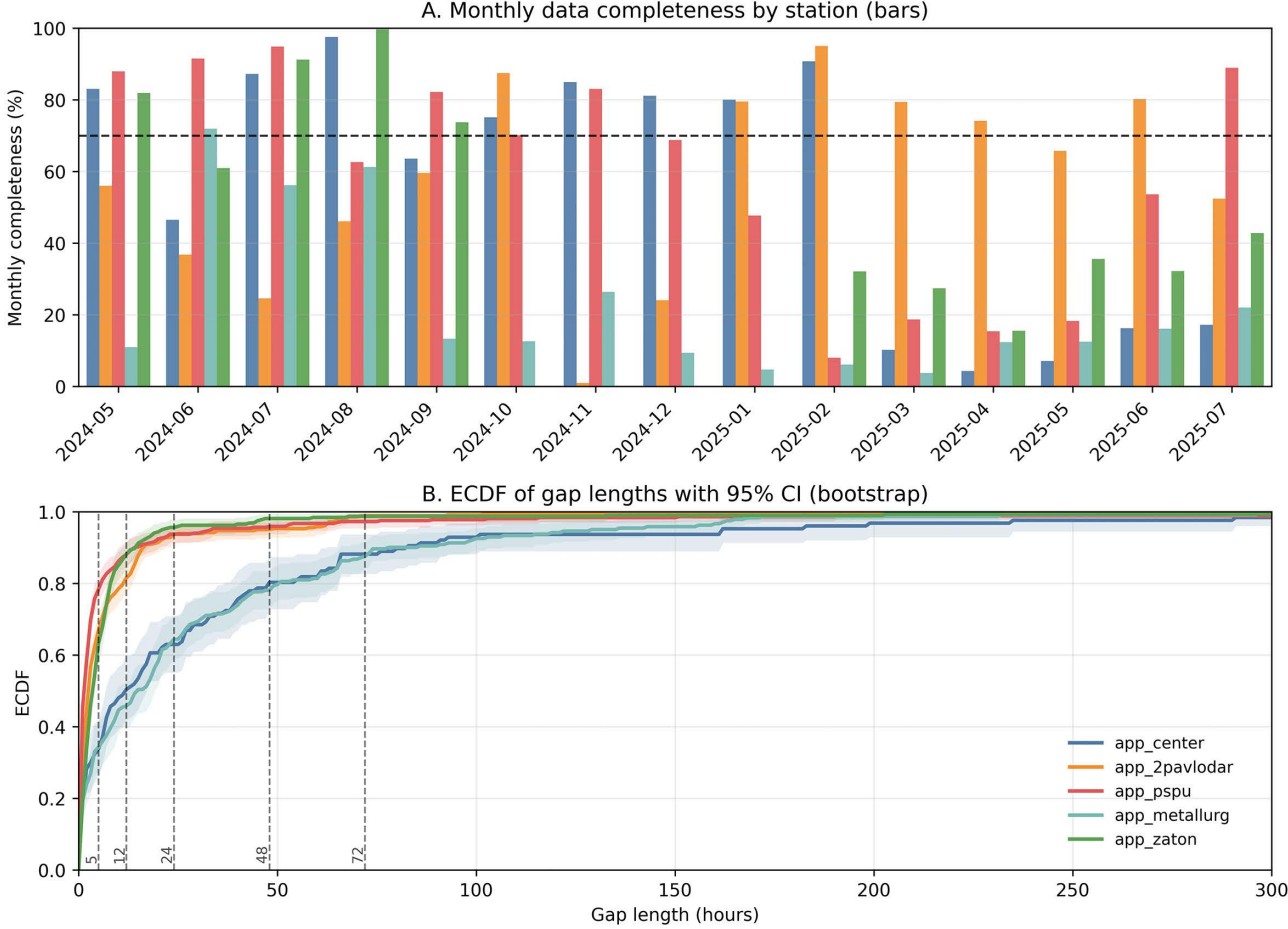

**Fig 4. Temporal patterns of data completeness and gap distribution across monitoring stations.** Panel A: Monthly data completeness rates for each monitoring station with 70% regulatory threshold indicated by horizontal dashed line. Panel B: Empirical cumulative distribution functions (ECDF) of gap lengths with 95% confidence intervals derived from bootstrap resampling of gap episodes. Critical thresholds marked at 5, 12, 24, 48, and 72 hours correspond to operational breakpoints for imputation strategy selection. Each station exhibits distinct gap duration profiles, with steeper curves indicating predominance of shorter interruptions and gradual slopes reflecting higher proportions of extended outages.

quality. After selective compression, remaining gaps were substantially shorter (mean: 3.5–12.7 hours vs. original 11.7–35 hours), facilitating more effective imputation. Full compression maintained 100% data retention with universal applicability across all monitoring stations.

Pollution concentration patterns varied considerably across stations, with mean PM$_{2.5}$ concentrations ranging from 17.5 µg/m³ (app_zaton) to 84.8 µg/m³ (app_metallurg). After selective compression, retained high-quality data showed modified concentration profiles, suggesting potential systematic relationships between data availability and pollution episodes that warrant consideration in imputation strategy selection (Table 5).

A brief comparative analysis of station-specific outcomes shows that the stations with more stable temporal structure and moderate variability in PM2.5 (app_center, app_pspu) benefited most from selective compression, which provides longer uninterrupted segments for model training. In contrast, stations with irregular emission patterns or lower initial completeness (app_zaton, app_2pavlodar) showed larger improvements under full compression, likely because this strategy

**Table 5. Data quality characteristics and compression strategy outcomes for the Pavlodar monitoring network.**

| Parameter | app_center | app_2pavlodar | app_pspu | app_metallurg | app_zaton |
|---|---|---|---|---|---|
| Original Completeness (%) | 56 | 57.5 | 57.4 | 23.3 | 37.4 |
| Original Gaps Number | 127 | 363 | 370 | 242 | 321 |
| Original Mean Gap (hours) | 35 | 11.9 | 11.7 | 32.1 | 19.8 |
| Original Max Gap (hours) | 340 | 1253 | 773 | 307 | 3268 |
| Original Mean PM$_{2.5}$ (µg/m³) | 25.1 | 33.8 | 64.3 | 84.8 | 17.5 |
| Original Max PM$_{2.5}$ (µg/m³) | 159.2 | 279.4 | 880.6 | 896.8 | 255.5 |
| Filtered Completeness (%) | 85.1 | 82.5 | 85 | 0 | 87.8 |
| Filtered Gaps Number | 62 | 157 | 184 | 0 | 38 |
| Filtered Mean Gap (hours) | 12.7 | 4.8 | 3.5 | 0 | 7.8 |
| Filtered Max Gap (hours) | 162 | 93 | 66 | 0 | 54 |
| Filtered Mean PM$_{2.5}$ (µg/m³) | 28.1 | 44.4 | 72.1 | 0 | 12.4 |
| Filtered Max PM$_{2.5}$ (µg/m³) | 159.2 | 279.4 | 880.6 | 0 | 208.2 |
| Selective Compression Ratio | 0.524 | 0.428 | 0.425 | 0 | 0.239 |

**Note:** Completeness represents percentage of valid hourly measurements over the 15-month study period (May 2024 – July 2025). Gap statistics calculated for consecutive missing periods ≥1 hour. Compression ratios indicate proportion of data retained under selective and full compression strategies. Selective compression employed 70% monthly completeness thresholds, minimum 24 consecutive valid hours, and minimum 3 months above the 70% completeness threshold. All stations achieved 100% retention under full compression strategy.

preserves all available valid observations, maximizing the usable signal in environments with higher variability or operational interruptions. The app_metallurg station, characterized by extremely sparse measurements, benefited exclusively from full compression, as selective filtering removed the station entirely from training.

### 4.2. Benchmarking performance under full compression strategy

Comprehensive evaluation of DynamicSeq2SeqXGB against standard imputation methods under full compression conditions demonstrated near-universal superiority across the monitoring network. In all experiments, DynamicSeq2SeqXGB was trained on multivariate input that combined PM$_{2.5}$ with meteorological covariates (temperature, humidity, wind speed and direction), which enhances the environmental interpretability of the imputed series. The benchmarking analysis, conducted across 25 gap length-station combinations, revealed consistent outperformance of the proposed machine learning approach, with 96% win rate (24/25 cases) and improvements averaging 43.4% over the best baseline methods (Table 6). Fig 5 shows percentage improvement in MAE for each station and gap length combination.

Performance overview established DynamicSeq2SeqXGB as the dominant imputation approach, achieving superiority in all cases except one instance at app_pspu for 48-hour gaps, where linear interpolation proved marginally superior (–28.3%). The consistent advantage across diverse operational conditions validates the effectiveness of the sequence-to-sequence architecture with adaptive context processing for environmental time series reconstruction under real-world monitoring constraints.

Station-specific analysis revealed distinct performance profiles while maintaining universal superiority. app_2pavlodar demonstrated the highest average improvements (52.0%), with peak performance reaching 64.6% for 24-hour gaps compared to rolling mean methods. app_zaton achieved substantial gains (50.2% average), with notable 60.0% improvement for 24-hour gaps over linear interpolation. app_metallurg showed the most variable performance (47.9% average), including the maximum single improvement of 80.9% for 72-hour gaps, while app_center (37.9% average) and app_pspu (29.0% average) demonstrated more moderate but consistent advantages.

**Table 6. DynamicSeq2SeqXGB performance comparison with baseline methods under full compression.**

| Station | Gap Length (hours) | DynamicSeq2SeqXGB MAE | Best Baseline MAE | Best Baseline Method | Improvement (%) |
|---|---|---|---|---|---|
| **app_2pavlodar** | 5 | **7.39±0.91** | 8.53±0.00 | Linear Interpolation | 13.3 |
| | 12 | **6.15±0.61** | 15.35±4.05 | Linear Interpolation | 59.9 |
| | 24 | **6.79±0.85** | 19.16±4.71 | Rolling Mean | 64.6 |
| | 48 | **7.78±0.97** | 20.58±4.22 | Rolling Mean | 62.2 |
| | 72 | **8.52±0.95** | 21.20±5.93 | Rolling Mean | 59.8 |
| **app_center** | 5 | **5.64±0.95** | 8.82±0.00 | Linear Interpolation | 36.1 |
| | 12 | **5.84±1.36** | 10.42±1.34 | Linear Interpolation | 43.9 |
| | 24 | **6.78±1.39** | 10.76±2.04 | Linear Interpolation | 37.0 |
| | 48 | **7.94±3.86** | 11.24±3.84 | Linear Interpolation | 29.3 |
| | 72 | **6.58±2.95** | 11.58±5.30 | Rolling Mean | 43.2 |
| **app_metallurg** | 5 | **25.61±8.16** | 43.98±16.73 | Forward Fill | 41.8 |
| | 12 | **31.14±19.81** | 42.28±24.22 | Linear Interpolation | 26.3 |
| | 24 | **27.18±26.48** | 42.57±31.76 | Linear Interpolation | 36.2 |
| | 48 | **29.71±37.17** | 65.00±60.24 | Linear Interpolation | 54.3 |
| | 72 | **32.23±42.85** | 168.34±155.96 | Linear Interpolation | 80.9 |
| **app_pspu** | 5 | **27.62±3.77** | 53.08±14.31 | Forward Fill | 48.0 |
| | 12 | **24.85±5.47** | 49.11±14.23 | Linear Interpolation | 49.4 |
| | 24 | **27.41±5.24** | 62.14±18.48 | Linear Interpolation | 55.9 |
| | 48 | 39.20±12.61 | **30.56±21.66** | Linear Interpolation | −28.3 |
| | 72 | **31.21±11.36** | 39.12±33.41 | Backward Fill | 20.2 |
| **app_zaton** | 5 | **4.08±1.06** | 5.40±0.00 | Linear Interpolation | 24.4 |
| | 12 | **3.52±0.60** | 8.78±2.90 | Linear Interpolation | 59.9 |
| | 24 | **3.67±1.11** | 9.18±2.61 | Linear Interpolation | 60.0 |
| | 48 | **4.50±1.21** | 8.73±3.24 | Rolling Mean | 48.4 |
| | 72 | **4.52±1.74** | 10.82±5.35 | Linear Interpolation | 58.3 |

**Note:** MAE values ($\mu g/m^3$) represent mean±standard deviation across 10 experimental runs with identical synthetic gap placement. Bold values indicate superior performance for each station-gap combination. Improvement percentages calculated as (baseline_MAE – dynamic_MAE)/baseline_MAE×100%.

Gap length dependencies exhibited clear temporal patterns favoring extended reconstructions. Extended gaps (72 hours) showed the highest average improvements (52.5%), with DynamicSeq2SeqXGB excelling in complex multi-day pattern reconstruction. Medium-term gaps (24 hours) achieved strong performance (50.7% average), effectively capturing diurnal cycle dynamics. 12-hour gaps demonstrated substantial improvements (47.9% average), while 48-hour gaps (33.2% average) and 5-hour gaps (32.7% average) showed more modest but still significant advantages.

Baseline method comparison revealed linear interpolation as the most competitive conventional approach, serving as the best baseline in the majority of cases. This dominance reflects linear interpolation's effectiveness for capturing temporal trends across diverse gap lengths. Rolling mean methods proved competitive for medium to extended gaps at specific stations, while forward and backward fill methods achieved best performance primarily for short-term interruptions or specific station characteristics.

Performance magnitude analysis demonstrated substantial and practically significant advantages across the network. Improvements exceeded 50% in 40% of cases (10/25), indicating transformative reconstruction quality under optimal conditions. Five cases achieved improvements >60%, with app_metallurg's 72-hour reconstruction representing the peak performance at 80.9% improvement over linear interpolation. The single case of underperformance (−28.3% at app_pspu,

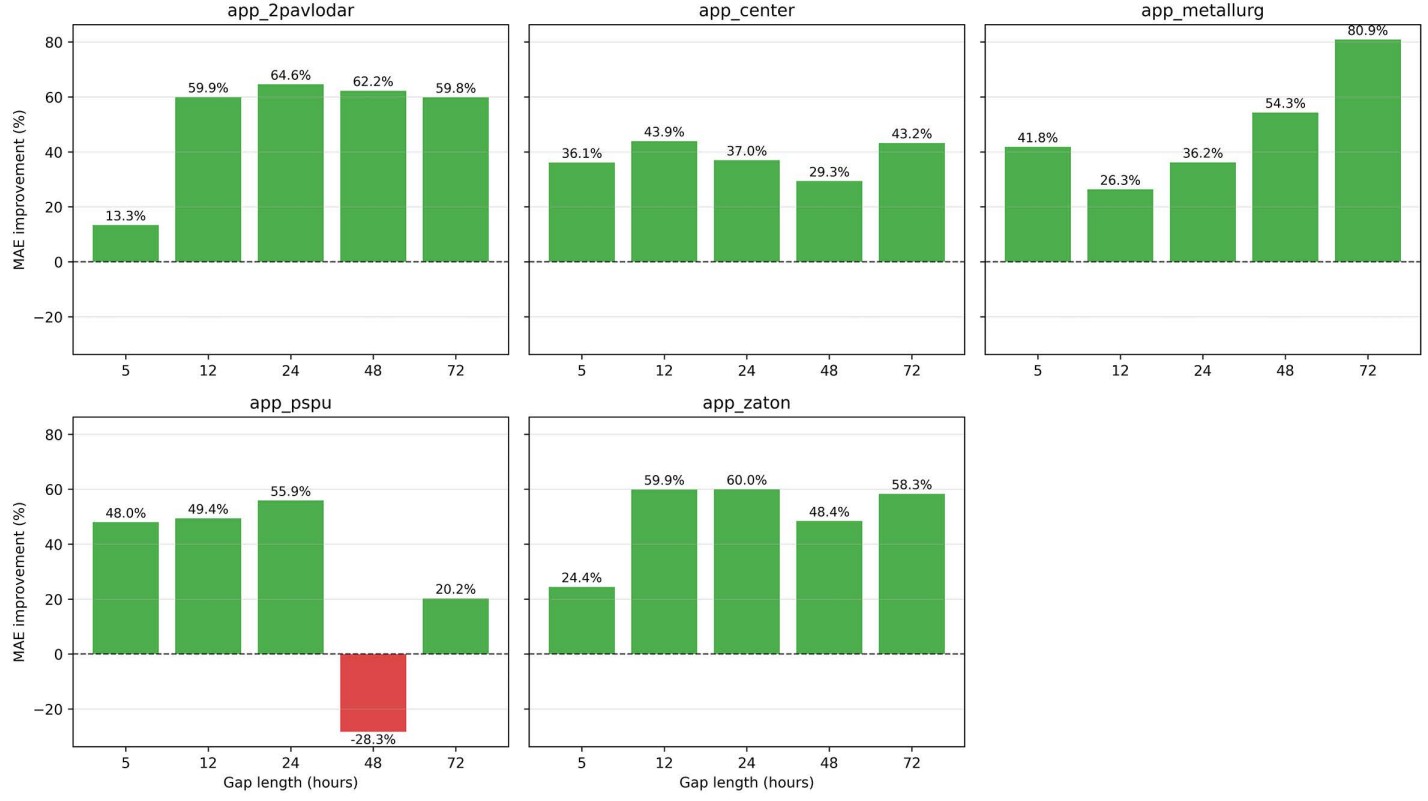

**Fig 5. DynamicSeq2SeqXGB improvement over best baseline methods under full compression strategy.** Green bars indicate DynamicSeq2Se-qXGB superiority; red bar indicates baseline method superiority. Horizontal dashed line at 0% represents performance parity.

48h) involved comparison with linear interpolation, suggesting that specific temporal patterns may occasionally favor simple trend-based approaches.

Methodological insights highlight the value of adaptive sequence-to-sequence processing for environmental time series reconstruction. The algorithm's superior performance on extended gaps demonstrates its ability to maintain temporal coherence across multiple diurnal cycles, where conventional methods struggle with pattern degradation. The predominance of linear interpolation as the primary competitor underscores the challenge of beating well-established temporal trend estimation, making DynamicSeq2SeqXGB's consistent superiority particularly noteworthy.

These results establish DynamicSeq2SeqXGB as a robust and superior alternative to conventional imputation methods for operational air quality monitoring networks. The near-universal improvements, particularly for challenging extended gaps, demonstrate that advanced machine learning approaches can deliver substantial and consistent value over traditional statistical methods when properly designed for environmental time series characteristics.

### 4.3. Benchmarking Performance under Selective Compression Strategy

Evaluation of DynamicSeq2SeqXGB under selective compression conditions revealed consistent superiority over classical imputation baselines, with the method achieving a 100% win rate (20/20 cases) and average improvements of 48.8% compared to the best baseline methods (Table 7). The analysis encompassed all stations except app_metallurg, which was excluded due to insufficient data quality for selective compression (compression ratio = 0.051).

**Table 7. DynamicSeq2SeqXGB performance comparison with baseline methods under selective compression.**

| Station | Gap Length (hours) | DynamicSeq2SeqXGB MAE | Best Baseline MAE | Best Baseline Method | Improvement (%) |
|---|---|---|---|---|---|
| **app_2pavlodar** | 5 | **7.98±0.65** | 15.61±0.00 | Linear Interpolation | 48.9 |
| | 12 | **7.35±1.15** | 22.19±5.67 | Linear Interpolation | 66.9 |
| | 24 | **9.30±2.38** | 19.27±4.14 | Rolling Mean | 51.7 |
| | 48 | **12.00±5.72** | 25.01±8.45 | Rolling Mean | 52 |
| | 72 | **12.18±4.66** | 20.48±5.18 | Rolling Mean | 40.5 |
| **app_center** | 5 | **5.55±0.73** | 6.64±0.00 | Linear Interpolation | 16.3 |
| | 12 | **5.74±1.35** | 12.46±2.81 | Linear Interpolation | 53.9 |
| | 24 | **7.63±1.66** | 12.87±1.98 | Linear Interpolation | 40.7 |
| | 48 | **6.28±1.97** | 15.92±6.28 | Linear Interpolation | 60.6 |
| | 72 | **10.26±5.06** | 17.17±7.45 | Linear Interpolation | 40.3 |
| **app_pspu** | 5 | **23.55±5.21** | 34.46±0.00 | Linear Interpolation | 31.6 |
| | 12 | **21.39±5.77** | 60.87±35.24 | Linear Interpolation | 64.9 |
| | 24 | **19.43±4.26** | 70.86±29.33 | Linear Interpolation | 72.6 |
| | 48 | **32.57±13.11** | 59.77±37.76 | Linear Interpolation | 45.5 |
| | 72 | **26.52±10.79** | 63.98±52.25 | Forward Fill | 58.6 |
| **app_zaton** | 5 | **3.02±1.16** | 5.98±0.29 | Linear Interpolation | 49.4 |
| | 12 | **3.95±1.81** | 7.72±2.25 | Rolling Mean | 48.8 |
| | 24 | **5.45±3.38** | 9.11±3.19 | Rolling Mean | 40.2 |
| | 48 | **3.31±4.43** | 7.35±2.37 | Rolling Mean | 55 |
| | 72 | **6.03±5.47** | 9.56±6.03 | Rolling Mean | 36.9 |

**Note:** MAE values ($\mu g/m^3$) represent mean±standard deviation across 10 experimental runs. Bold values indicate DynamicSeq2SeqXGB performance (superior in all cases). Improvement percentages calculated as (baseline_MAE – dynamic_MAE)/baseline_MAE×100%. app_metallurg excluded due to insufficient data quality for selective compression.

Performance overview established DynamicSeq2SeqXGB's universal dominance under selective compression, achieving superiority in all evaluated cases without exception. This perfect win rate contrasts with the 96% success rate observed under full compression, suggesting that selective preprocessing may enhance the method's robustness across diverse operational conditions. The consistent advantage validates the effectiveness of quality-first data preparation strategies for environmental time series reconstruction.

Station-specific analysis revealed distinct performance profiles with all stations benefiting from selective compression. app_pspu demonstrated the highest average improvements (54.6%), with exceptional performance for 24-hour gaps (72.6% improvement over linear interpolation) and 12-hour gaps (64.9% improvement over linear interpolation). app_2pavlodar achieved substantial gains (52.0% average) with peak performance of 66.9% for 12-hour gaps, while app_center showed consistent improvements (42.4% average) across all gap lengths. app_zaton exhibited strong performance (46.1% average) with notable 55.0% improvement for 48-hour gaps over rolling mean methods.

Gap length dependencies under selective compression displayed favorable patterns across all temporal scales. 12-hour gaps demonstrated the strongest average improvements (58.6%), effectively capturing sub-daily cycle dynamics through enhanced data quality. 48-hour gaps showed strong performance (53.3% average), while 24-hour gaps achieved substantial improvements (51.3% average). 72-hour gaps (44.1% average) and 5-hour gaps (36.6% average) exhibited more modest but still significant advantages.

Baseline method comparison revealed linear interpolation as the predominant competitor under selective compression, serving as the best baseline in 65% of cases. This dominance reflects the method's effectiveness when applied to

quality-filtered data with enhanced temporal coherence. Rolling mean and backward fill methods achieved competitive performance for specific station-gap combinations, while forward fill proved optimal primarily for short-term interruptions.

Performance magnitude analysis demonstrated substantial improvements across the network. Improvements exceeded 50% in 45% of cases (9/20), with the maximum improvement reaching 72.6% for app_pspu's 24-hour reconstruction. The minimum improvement of 16.3% occurred for app_center's 5-hour gaps, indicating that even in less favorable scenarios, the method maintained consistent advantages over conventional approaches.

Methodological insights highlight the value of selective compression for enhancing reconstruction quality. The universal success rate suggests that quality-first preprocessing strategies create more favorable conditions for sequence-to-sequence learning by reducing noise and improving temporal pattern consistency. The particularly strong performance on short to medium-duration gaps (12–48 hours) reflects the method's ability to leverage enhanced data quality for capturing complex atmospheric dynamics.

These results establish that selective compression enhances DynamicSeq2SeqXGB's robustness across diverse monitoring conditions, achieving universal superiority over conventional methods. The findings support the implementation of adaptive preprocessing strategies that prioritize data quality when sufficient historical observations are available for effective model training.

## 4.4. Comparative analysis of compression strategies

Systematic comparison between full and selective compression strategies revealed distinct performance characteristics with implications for operational deployment in environmental monitoring networks. The analysis, conducted across 20 comparable station-gap combinations (excluding app_metallurg due to selective compression incompatibility), demonstrated that strategy selection significantly impacts reconstruction quality in a context-dependent manner.

Overall performance patterns showed both compression strategies achieving universal superiority over classical baseline methods when evaluated independently. However, direct head-to-head comparison revealed balanced competition with equal win rates – full compression achieving superiority in 50% of cases (10/20) versus 50% for selective compression (10/20). The comparative analysis provides critical insights into the relative effectiveness of quality-first versus quantity-first data preparation methodologies for machine learning-based imputation.

Strategy effectiveness patterns varied substantially across temporal scales and station characteristics (Table 8). For short-term gaps (5 hours), selective compression demonstrated advantages in 75% of comparable cases (3/4), suggesting that quality-filtered data provides superior context for brief interruptions. Medium-term gaps showed mixed patterns: 12-hour gaps exhibited balanced performance (50% each), while 24-hour gaps favored full compression in 75% of cases (3/4). Extended gaps displayed divergent patterns, with 48-hour gaps favoring selective compression (75%) while 72-hour gaps preferred full compression (75%).

Station-specific preferences emerged as the dominant performance determinant and are more pronounced than any gap-length-based trends, as shown in Fig 6. app_pspu consistently benefited from selective compression across all gap durations, with selective improvements ranging from 16% to 41%, indicating that strict quality filtering preserves temporally coherent structure essential for reconstruction at this station. In contrast, app_2pavlodar showed uniform preference for full compression, with performance gains between 8% and 54%, reflecting the benefits of retaining all available observations at a station characterized by frequent interruptions and irregular measurement patterns. app_zaton also demonstrated an overall tendency toward full compression – particularly for medium and long gaps (12–24 hours and 72 hours) – though selective compression remained advantageous for the shortest (5 h) and 48-hour gaps, indicating mixed temporal behavior. app_center displayed a hybrid pattern: selective compression provided small improvements for short interruptions, whereas full compression delivered the largest single-station gain in the dataset (+56% at 72 hours). These patterns illustrate that environmental variability, sensor stability, and data continuity differ substantially across stations and exert stronger influence on strategy effectiveness than gap duration alone.

**Table 8. Direct comparison of compression strategies: MAE values and strategy superiority.**

| Station | Gap Length (hours) | Full Compression MAE | Selective Compression MAE | Superior Strategy | Improvement (%) |
|---|---|---|---|---|---|
| app_center | 5 | 5.63 | 5.55 | Selective | 1.5 |
| app_center | 12 | 5.84 | 5.74 | Selective | 1.8 |
| app_center | 24 | 6.78 | 7.63 | Full | 12.6 |
| app_center | 48 | 7.94 | 6.28 | Selective | 26.5 |
| app_center | 72 | 6.58 | 10.26 | Full | 56 |
| app_2pavlodar | 5 | 7.39 | 7.98 | Full | 8 |
| app_2pavlodar | 12 | 6.15 | 7.35 | Full | 19.5 |
| app_2pavlodar | 24 | 6.79 | 9.3 | Full | 37 |
| app_2pavlodar | 48 | 7.78 | 12 | Full | 54.3 |
| app_2pavlodar | 72 | 8.52 | 12.18 | Full | 43 |
| app_pspu | 5 | 27.62 | 23.55 | Selective | 17.3 |
| app_pspu | 12 | 24.85 | 21.39 | Selective | 16.2 |
| app_pspu | 24 | 27.41 | 19.43 | Selective | 41.1 |
| app_pspu | 48 | 39.2 | 32.57 | Selective | 20.4 |
| app_pspu | 72 | 31.21 | 26.52 | Selective | 17.7 |
| app_zaton | 5 | 4.08 | 3.02 | Selective | 35 |
| app_zaton | 12 | 3.52 | 3.95 | Full | 12.2 |
| app_zaton | 24 | 3.67 | 5.45 | Full | 48.4 |
| app_zaton | 48 | 4.5 | 3.31 | Selective | 36.1 |
| app_zaton | 72 | 4.52 | 6.03 | Full | 33.5 |
| **Summary** | | **Full wins: 10/20 (50%)** | **Selective wins: 10/20 (50%)** | | |

**Note:** MAE values ($\mu g/m^3$) represent mean across 10 experimental runs. Bold values indicate superior performance for each comparison. Improvement percentages calculated as (inferior_MAE – superior_MAE)/superior_MAE × 100%, where inferior_MAE denotes the numerically larger MAE value (indicating worse performance) and superior_MAE denotes the numerically smaller MAE value (indicating better performance), representing the relative performance gain of the superior strategy over the inferior one. Analysis excludes app_metallurg due to incompatibility with selective compression.

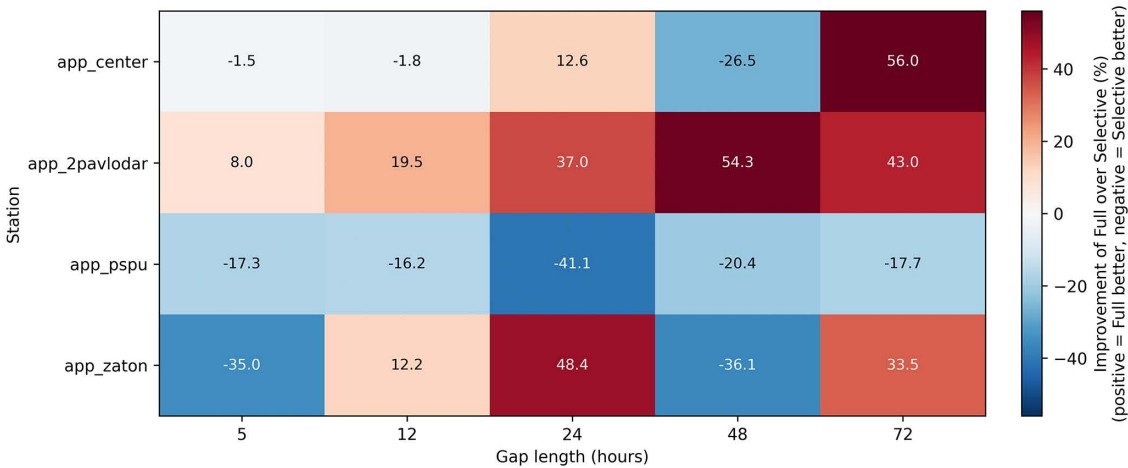

**Fig 6. Strategy performance advantage by station and gap length.** Heatmap showing percentage improvement of full compression over selective compression. Positive values (orange/red) indicate full compression advantage; negative values (blue) indicate selective compression advantage. Clear horizontal patterns demonstrate station-specific preferences that exceed gap-length-based trends, with app_pspu showing consistent preference for selective compression while app_2pavlodar favors full compression across all gap lengths.

Direct comparison analysis across matched station-gap combinations provides quantitative evidence for strategy selection patterns (Table 8). The head-to-head comparison revealed perfectly balanced overall performance, with average improvements of 28.6% when the superior strategy was applied over the inferior alternative. However, pronounced station-specific variations indicate that universal strategy recommendations are inappropriate for operational deployment.

Performance magnitude assessment demonstrated substantial differences between strategies in optimal application scenarios. The largest observed improvement occurred at app_center for 72-hour gaps, where full compression achieved 56.0% superior performance over selective compression. The peak advantage for selective compression reached 48.4% at app_zaton for 24-hour gaps. These substantial differences underscore the critical importance of appropriate strategy selection for maximizing reconstruction quality.

Gap length dependency analysis revealed complex relationships between temporal scale and strategy effectiveness (Fig 7). Linear trend analysis shows median MAE for selective compression increases with gap length, while full compression maintains relatively stable performance across temporal scales. The relative performance gap between strategies varied significantly, with individual data points demonstrating substantial station-specific variations around the fitted trends, confirming that neither strategy shows consistent dominance across all gap lengths.

Critical operational differences emerged in methodological robustness and universal applicability. Full compression maintained universal applicability across all monitoring stations, while selective compression proved incompatible with severely compromised stations (app_metallurg elimination). This distinction has profound implications for operational networks requiring consistent methodology across heterogeneous station types, where strategy failures could create critical monitoring blind spots.

Methodological insights highlight the fundamental trade-off between data quality and data quantity in machine learning applications. Selective compression's quality-first approach creates more favorable conditions for temporal pattern recognition when sufficient high-quality data is available, as evidenced by app_pspu's consistent preference. However, full compression's quantity-first methodology proves more robust across diverse data quality scenarios, particularly for stations with moderate data quality issues where additional context outweighs quality concerns.

These findings establish that strategy selection should be informed by station-specific characteristics rather than universal temporal thresholds. The strong consistency of station preferences (app_pspu and app_2pavlodar showing 100%

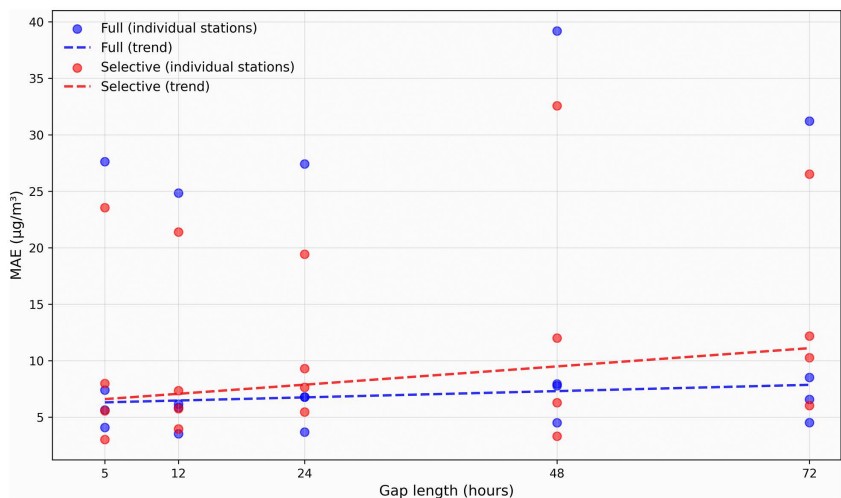

**Fig 7. Imputation error (MAE) as a function of gap length under different compression strategies.** Linear trends show median MAE across four stations (excluding app_metallurg) for selective (red) and full (blue) compression approaches. Individual data points represent MAE values for each station at different gap lengths.

preference for their respective optimal strategies) indicates that local data quality profiles, sensor characteristics, and environmental conditions fundamentally influence reconstruction effectiveness more than temporal gap properties alone. For operational deployment, this suggests implementing adaptive strategy selection protocols based on historical station performance profiles rather than gap-length-based decision trees.

Notably, both compression strategies demonstrated substantial peak performance advantages in their respective optimal scenarios (56.0% for full compression, 48.4% for selective compression), suggesting that strategy selection potential is significant regardless of the chosen approach, with station-specific characteristics being the determining factor for optimal performance rather than inherent strategy superiority.

### 4.5. Correlation Analysis of Station Characteristics and Compression Strategy Performance

To understand the factors underlying compression strategy effectiveness, Pearson correlation analysis was conducted across nine variables derived from the four-station monitoring network (excluding app_metallurg due to selective compression incompatibility). The analysis examined relationships between station characteristics and compression strategy performance, where Improvement_Full_vs_Selective_% represents the percentage performance gain of full compression relative to selective compression, Homogeneity quantifies temporal consistency in $PM_{2.5}$ measurements, Mean_PM25 and Std_PM25 denote average concentration and temporal variability respectively, and Missing_Count indicates total missing observations per station.

The correlation matrix revealed seven statistically significant relationships out of 36 possible pairwise correlations ($p < 0.05$), providing quantitative insight into compression strategy effectiveness patterns. The most striking finding was the perfect negative correlation between data completeness and missing count (Completeness_% ↔ Missing_Count: $r = -1.000$, $p < 0.001$), confirming the mathematical relationship between these complementary measures of data availability.

Strong positive correlations emerged between pollution characteristics and reconstruction errors for both compression strategies. Full compression MAE showed near-perfect correlation with $PM_{2.5}$ temporal variability (Full_Compression_MAE ↔ Std_PM25: $r = 0.997$, $p = 0.003$), while selective compression MAE correlated strongly with mean pollution levels (Selective_Compression_MAE ↔ Mean_PM25: $r = 0.996$, $p = 0.004$). These relationships indicate that both strategies encounter greater challenges at stations with high pollution concentrations and temporal variability, reflecting the inherent difficulty of reconstructing complex atmospheric patterns.

The analysis revealed that strategy improvement patterns are driven primarily by temporal consistency rather than data availability. Improvement_Full_vs_Selective_% showed the strongest correlation with Homogeneity ($r = 0.886$), indicating that full compression performs particularly well at stations with consistent temporal patterns. Conversely, stations with high temporal variability (Std_PM25: $r = -0.86$) and reconstruction errors (Full_Compression_MAE: $r = -0.859$) showed reduced advantages for full compression, though these correlations did not reach statistical significance ($p > 0.05$).

Station-specific analysis revealed distinct pollution profiles that explain strategy preferences. app_pspu exhibited the highest mean $PM_{2.5}$ concentrations (64.3 µg/m³) with extreme temporal variability ($\sigma = 161.3$ µg/m³), while app_center and app_zaton demonstrated lower pollution levels (25.1 and 17.5 µg/m³ respectively) with moderate variability. app_2pavlodar occupied an intermediate position with moderate mean concentrations (33.8 µg/m³) and variability ($\sigma = 38.6$ µg/m³). These pollution characteristics align with the observed station-specific strategy preferences, where high-variability stations (app_pspu) consistently favor selective compression while more stable stations (app_2pavlodar) benefit from full compression's additional contextual information.

The strong correlation between pollution levels and temporal variability (Mean_PM25 ↔ Std_PM25: $r = 0.967$, $p = 0.033$) confirms that high-pollution stations present inherently more complex reconstruction challenges. Both compression strategies show near-perfect correlation in their error patterns (Full_Compression_MAE ↔ Selective_Compression_MAE: $r = 0.990$, $p = 0.010$), indicating that while their absolute performance may differ, they encounter similar challenges at problematic stations.

These correlational patterns provide empirical evidence that strategy selection should be based on station pollution characteristics rather than data availability metrics. The lack of significant correlation between completeness and strategy improvement (Improvement_Full_vs_Selective_% ↔ Completeness_%: r = −0.087, p = 0.913) demonstrates that data quality filtering effectiveness is independent of baseline data availability, supporting adaptive strategy selection protocols based on pollution variability and temporal consistency rather than simple completeness thresholds. Complete correlation coefficients and significance levels are provided in Supplementary Table S3 in S1 File.

## 4.6. External Validation (Beijing, Guanyuan, 2016)

To assess the transferability and robustness of DynamicSeq2SeqXGB beyond the training environment, external validation was conducted using high-quality reference data from the Beijing PRSA dataset (Guanyuan station, 2016, https://doi.org/10.24432/C5RK5G) [54]. This cross-regional evaluation employed the full compression strategy exclusively, providing an independent assessment of the method's performance across different geographical, climatic, and pollution characteristics. The validation used a controlled degradation approach to simulate realistic monitoring network failures with large gaps comparable to those observed in the Pavlodar network, examining whether patterns learned from Central Asian monitoring conditions remain effective in East Asian urban environments with distinct atmospheric dynamics and pollution sources.

The Beijing Guanyuan dataset for 2016 provided an optimal validation framework with 8,784 hourly observations and 98.41% initial completeness. A complex degradation pattern was systematically applied based on gap characteristics observed in the Pavlodar app_pspu station, combining long clustered outages with scattered single-hour missing values to realistically reproduce its empirical sparsity profile.

To ensure reproducibility and clarify the construction of this missingness pattern, we explicitly mirrored the empirical gap structure of the Pavlodar app_pspu station. First, we extracted the six longest real outage episodes from app_pspu and inserted gaps of identical durations (773, 383, 360, 193, 180, and 95 hours) at randomly selected positions in the Beijing time series, thereby reproducing its characteristic long clustered failures. We then added 2,408 single-hour missing values sampled uniformly at random across the remaining valid timestamps to capture the short, dispersed interruptions typical of this station. This procedure closely matches the observed gap-length distribution of app_pspu and produces a complex missingness profile combining long clustered outages and dispersed single-point gaps (visualized in Supplementary Fig S5 in S1 File). Because the objective of this experiment was to evaluate robustness under a realistic worst-case sparsity profile derived from an operationally degraded station, we did not incorporate additional synthetic missingness schemes such as purely random, seasonal, or independently clustered gaps. This controlled degradation reduced data completeness from 98.41% to 54.64%, creating 3,844 artificially missing observations for evaluation while preserving 4,800 dense data points for model training.

Following the established methodology, the DynamicSeq2SeqXGB model was trained exclusively on the 4,800 dense data points using air quality parameters ($PM_{10}$, $SO_2$, $NO_2$, $CO$, $O_3$), meteorological variables (temperature, pressure, dew point, precipitation, wind speed), and temporal features (hour, day of week, month). The XGBoost regressor employed 100 estimators with maximum depth of 6 and StandardScaler normalization.

The external validation demonstrated exceptional performance across all evaluation metrics, confirming the model's ability to maintain high reconstruction quality under challenging operational conditions (Table 9).

The strong linear relationship between predicted and actual values is clearly demonstrated in the scatter plot analysis (Fig 8), which shows excellent agreement with the ideal prediction line and confirms the absence of systematic bias in model predictions.

The DynamicSeq2SeqXGB model achieved MAE of 8.50 μg/m³, RMSE of 18.84 μg/m³, and $R^2$ of 0.944, representing substantial improvement over baseline methods across 3,844 validation points. Linear interpolation achieved only $R^2 = 0.303$, highlighting the critical advantage of the machine learning approach for complex temporal pattern

**Table 9. External validation performance on Beijing dataset (Guanyuan, 2016).**

| Method | MAE (µg/m³) | RMSE (µg/m³) | R² | Improvement (%) |
|---|---|---|---|---|
| **DynamicSeq2SeqXGB** | **8.5** | **18.84** | **0.944** | **68.1-79.8** |
| Linear Interpolation | 36.82 | 66.69 | 0.303 | 68.1 |
| Forward Fill | 37.68 | 66.38 | 0.310 | 68.8 |
| Backward Fill | 45.80 | 74.66 | 0.121 | 74.3 |
| Mean Imputation | 58.27 | 79.91 | 0.000 | 79.8 |
| Median Imputation | 55.46 | 84.02 | −0.106 | 78.8 |

**Note:** The Improvement column shows the percentage MAE reduction achieved by DynamicSeq2SeqXGB compared to each baseline method, calculated as (baseline_MAE – DynamicSeq2SeqXGB_MAE)/baseline_MAE × 100%. For classical methods, the values represent how much better DynamicSeq2SeqXGB performs relative to that specific method. The range 68.1–79.8% for DynamicSeq2SeqXGB indicates the span of improvements achieved over all baseline methods.

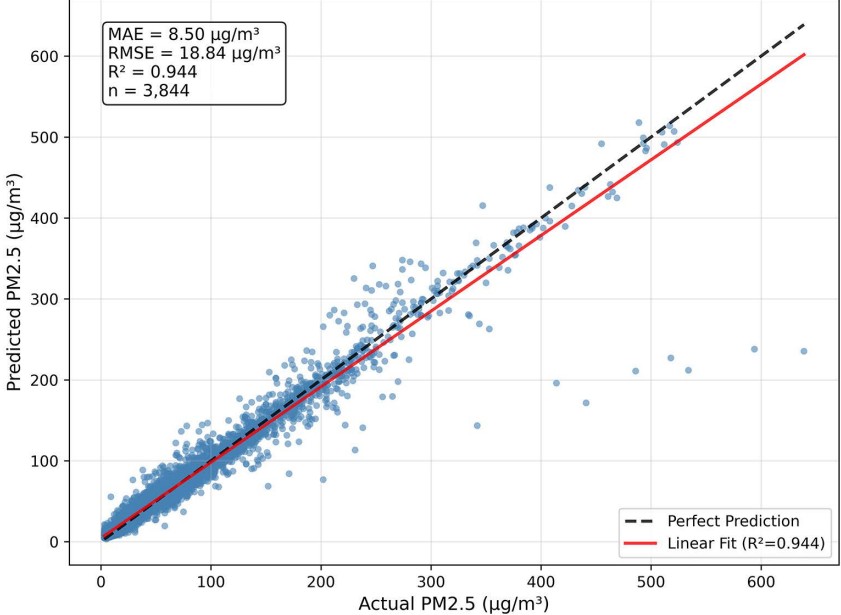

**Fig 8. External validation results showing predicted vs actual PM$_{2.5}$ concentrations and error distribution for DynamicSeq2SeqXGB model on Beijing dataset.** Panel A demonstrates strong linear agreement between predicted and actual values with R²=0.921, with the red line representing model fit and the black dashed line indicating perfect prediction (y=x).

reconstruction. The model maintained consistent performance across the entire concentration range (3.0–639.0 µg/m³) with minimal systematic bias (−0.58 µg/m³), demonstrating robust handling of both background and pollution episode conditions. Some tendency toward underestimation was observed at extreme pollution levels (>400 µg/m³, bias: −75.74 µg/m³ for 31 points), though this represents a small fraction of the validation dataset (0.8%) and does not significantly impact overall performance metrics.

Error behavior across concentration levels was additionally examined to evaluate whether reconstruction accuracy deteriorates during pollution episodes. The model maintained stable performance throughout most of the concentration range (0–300 µg/m³), with MAE varying moderately between 6–10 µg/m³. A clear increase in error was observed only at

the upper tail of the distribution: for concentrations above 400 µg/m³ (0.8% of observations), the model exhibited systematic underestimation with an average bias of −75.74 µg/m³. This pattern reflects the scarcity of extreme pollution events in the training data rather than structural model limitations. Importantly, the small number of such events means that these localized errors do not materially affect aggregate performance metrics (MAE, RMSE, $R^2$), which remain robust across the full concentration spectrum.

The results confirm robust cross-domain transferability, with the model trained on Central Asian data (Pavlodar) successfully generalizing to East Asian conditions (Beijing) despite different pollution sources and meteorological patterns. Performance remained stable across diverse gap configurations spanning from 48 to 773 hours, demonstrating scalability to operational monitoring challenges comparable to real-world network failures. The maintained 68.1–79.8% improvement margins over baseline methods across different geographic regions and pollutant concentration ranges provide compelling evidence that the approach captures fundamental atmospheric dynamics rather than domain-specific artifacts, supporting practical deployment in diverse air quality monitoring networks facing severe data limitations.

## 5. Discussion

### 5.1. Performance of DynamicSeq2SeqXGB and compression strategy selection

The consistently superior performance of DynamicSeq2SeqXGB highlights its ability to capture complex environmental dynamics and demonstrates clear advantages over conventional imputation methods. This near-universal superiority contrasts markedly with previous benchmarking studies where performance advantages typically ranged from 20–40% for specialized methods. For instance, while SAITS demonstrated 12–38% improvements over RNN-based approaches in controlled settings [21], and BRITS-ALSTM achieved $R^2 = 0.7454$ under extreme sparsity conditions [3], DynamicSeq2SeqXGB delivered consistent improvements averaging 48.8% across diverse operational conditions. The missForest algorithm, previously considered state-of-the-art for environmental data with 92.41% accuracy [39], is substantially outperformed by presented approach which maintains effectiveness even at 23.3% data completeness. Recent regional studies also demonstrate the competitive performance of gradient-boosted tree models for particulate matter prediction, including XGBoost-based $PM_{10}$ modeling in operational contexts [55].

The balanced performance between full and selective compression strategies (50% win rate each) reveals that optimal preprocessing depends critically on station-specific characteristics rather than universal quality thresholds. This finding challenges conventional data preprocessing paradigms that assume quality filtering consistently improves machine learning performance. The correlation analysis provides empirical evidence for this station-dependency, showing that strategy improvement patterns are driven primarily by temporal consistency (Homogeneity: $r = 0.886$) rather than data availability (Completeness: $r = −0.087$, $p = 0.913$).

Station-specific preferences demonstrated remarkable consistency, with app_pspu exclusively benefiting from selective compression across all gap lengths (improvements 16.2–41.1%) while app_2pavlodar consistently favored full compression (improvements 8.0–54.3%). This consistency suggests that local pollution characteristics and temporal patterns fundamentally influence optimal preprocessing strategies more than gap-specific properties.

The app_metallurg case exemplifies both the challenges and limitations of imputation under extreme sparsity. While full compression enabled reconstruction where selective compression failed entirely (due to 0% qualifying months), the resulting MAE values (29.71–32.23 µg/m³) were substantially higher than other stations (3.67–8.52 µg/m³), reflecting the inherent difficulty of accurate imputation at 23.3% completeness. This demonstrates that while DynamicSeq2SeqXGB can extract meaningful patterns from severely compromised datasets, reconstruction quality inevitably degrades as data sparsity increases.

The theoretical foundation for balanced strategy performance lies in fundamental trade-offs between data quality and quantity. Selective compression creates favorable conditions for pattern recognition by reducing noise, as evidenced by

app_pspu's consistent preference and strong correlation between pollution variability and reconstruction difficulty (Mean_PM25↔Std_PM25: r = 0.967, p = 0.033). Conversely, full compression preserves temporal dependencies that may contain essential reconstruction information, particularly beneficial for stations with moderate data quality where additional context outweighs noise concerns.

The framework's emphasis on station-specific strategy selection reflects operational priorities where consistent performance across heterogeneous conditions outweighs marginal accuracy gains in ideal scenarios. Rather than seeking universal optimization, the approach provides operators with clear, actionable guidance based on quantifiable station characteristics, simplifying deployment protocols while maintaining reconstruction effectiveness across diverse monitoring network conditions.

## 5.2. Theoretical implications of context-dependent strategy selection

The equal performance of full and selective compression strategies (50% win rate each) reveals fundamental principles about data preprocessing in extreme sparsity scenarios that merit theoretical consideration. Rather than validating universal preprocessing paradigms, these results demonstrate that optimal strategy selection depends on the complex interplay between data characteristics, temporal patterns, and reconstruction requirements.

The station-specific consistency of strategy preferences indicates that preprocessing effectiveness stems from alignment between data characteristics and algorithmic requirements rather than absolute data quality metrics. Correlation analysis supports this interpretation, showing that strategy improvement correlates with temporal consistency (Homogeneity: r = 0.886) while remaining independent of baseline data availability (Completeness: r = −0.087, p = 0.913). This pattern suggests that successful imputation depends on matching preprocessing approaches to underlying temporal structure rather than maximizing data cleanliness.

The theoretical foundation underlying these results centers on information preservation versus noise reduction trade-offs. Selective compression optimizes signal-to-noise ratios by eliminating low-quality observations, creating favorable conditions for pattern recognition when sufficient high-quality data exists. This approach proves particularly effective for stations with high pollution variability (app_pspu), where temporal inconsistency benefits from quality filtering. Conversely, full compression preserves temporal dependencies that may contain reconstruction-relevant information, particularly valuable when moderate noise levels are offset by increased contextual breadth.

These findings challenge the conventional assumption that preprocessing should universally optimize data quality before algorithmic application. Instead, the results suggest that preprocessing strategies should be viewed as integral components of the overall reconstruction system, where effectiveness depends on compatibility between data characteristics, preprocessing approach, and algorithmic architecture. The XGBoost ensemble's demonstrated ability to handle mixed-quality inputs enables this flexibility, suggesting that robust algorithms may benefit more from comprehensive information access than from pre-filtered datasets. Comparative evaluations of ensemble and boosting architectures for long-term air-quality forecasting further support the advantages of tree-based ensembles in handling heterogeneous temporal patterns [56].

The implications extend beyond environmental monitoring to broader questions about data preparation in machine learning applications. The station-specific nature of optimal strategies indicates that preprocessing decisions should incorporate domain knowledge about data generation processes, temporal dynamics, and measurement characteristics rather than relying solely on statistical quality metrics. This perspective shifts focus from universal best practices toward adaptive, context-aware preprocessing approaches that optimize system-level performance rather than individual component quality.

However, these theoretical implications must be interpreted within the study's limitations. The analysis encompasses only four stations with $PM_{2.5}$ data, limiting generalizability across different environmental parameters, geographic regions, and monitoring technologies. The observed patterns may reflect specific characteristics of atmospheric pollutant dynamics

rather than universal principles applicable to other domains with sparse time series data. Further research across diverse applications would be necessary to validate broader theoretical claims about context-dependent preprocessing optimization.

### 5.3. Practical implications for Environmental Monitoring Networks

The practical significance of these findings centers on optimizing data utilization within existing monitoring infrastructure constraints rather than transforming fundamental operational paradigms. The demonstrated station-specific strategy preferences provide actionable guidance for operators facing decisions about data preprocessing approaches, though the implications require careful interpretation within the study's empirical limitations.

The most immediate practical benefit lies in establishing evidence-based protocols for compression strategy selection. Stations exhibiting high pollution variability and temporal inconsistency (similar to app_pspu) benefit consistently from selective compression, while stations with moderate data quality and stable patterns (similar to app_2pavlodar) perform better under full compression. This station-specific approach contrasts with current practices that often apply uniform pre-processing protocols across entire networks, potentially suboptimizing reconstruction quality for individual sites.

The app_metallurg case illustrates both possibilities and limitations under extreme sparsity conditions. While full compression enabled reconstruction where selective compression failed entirely, the resulting reconstruction errors (MAE 29.71–32.23 µg/m$^3$) substantially exceeded those achieved at other stations (3.67–8.52 µg/m$^3$). Whether such reconstruction quality constitutes "meaningful" data depends critically on intended applications – these error levels might suffice for broad trend identification but would be inadequate for regulatory compliance or detailed exposure assessment. The cost-benefit analysis of maintaining severely compromised stations requires evaluation against specific operational requirements rather than assuming universal value.

The framework's emphasis on station-specific optimization reflects operational realities where heterogeneous infra-structure conditions necessitate adaptive rather than standardized approaches. However, this customization introduces implementation complexity, requiring operators to characterize station-specific temporal patterns and maintain differ-entiated preprocessing protocols. The additional analytical overhead must be weighed against potential reconstruction improvements when determining operational feasibility.

Regarding broader applicability to other environmental parameters, the current evidence base remains limited to PM$_{2.5}$ measurements from Central Asian monitoring conditions. While the methodology's reliance on temporal pattern recogni-tion suggests potential transferability to other continuously monitored pollutants (SO$_2$, NO$_2$, O$_3$), parameter-specific valida-tion would be essential given different measurement characteristics, temporal dynamics, and typical concentration ranges.

Beyond pollutant-specific considerations, applicability across different geographical contexts also warrants attention. Although the empirical analysis in this study is centered on Pavlodar, the DynamicSeq2SeqXGB framework is not tied to region-specific features. The model operates on general temporal properties of air-quality time series, including diur-nal structure, short-term temporal dependencies, and characteristic gap patterns, and relies only on routinely collected pollutant concentrations and basic meteorological variables. As a result, its applicability is primarily determined by data sparsity and the availability of core covariates rather than by local emission profiles. External validation on the high-quality Beijing Guanyuan PM2.5 time series, implemented through controlled degradation of an originally near-complete dataset, further supports this generality: despite substantial differences in climatic conditions and urban emission regimes between Beijing and Pavlodar, the model maintained high reconstruction accuracy. This indicates that the proposed approach can be transferred to other regions with comparable monitoring configurations, provided that similar temporal resolution and essential meteorological information are available.

The computational requirements of DynamicSeq2SeqXGB relative to classical methods represent another practical consideration. While the performance improvements often justify increased processing costs, resource-constrained moni-toring programs must evaluate this trade-off based on their specific accuracy requirements and computational capabilities.

The method's integration with existing data management systems would require careful planning to ensure sustainable implementation.

In practical terms, the computational burden of the proposed approach is modest. Because the encoder is based on gradient-boosted decision trees rather than deep recurrent or attention-based architectures, full model training for a single station typically completes within several minutes on a standard CPU (Intel i7–class). Once trained, inference is extremely lightweight: predictions for a given context–gap window are produced within milliseconds, enabling on-the-fly reconstruction without GPU resources or specialized hardware. This efficiency makes DynamicSeq2SeqXGB suitable for real-time or near–real-time deployment in municipal monitoring systems, where processing capacity is often limited and continuous data delivery is required.

The framework demonstrates most value for monitoring networks facing chronic data quality challenges where conventional approaches consistently underperform. For well-maintained networks with minimal data gaps, the added complexity may provide marginal benefits over simpler interpolation methods. The optimal application domain appears to be intermediate scenarios where data sparsity creates reconstruction challenges but sufficient temporal structure remains for machine learning approaches to extract meaningful patterns.

## 5.4. Study limitations and methodological considerations

Several considerations warrant discussion regarding the scope and generalizability of these findings. The correlation analysis, conducted across four monitoring stations, provides meaningful insights into station-specific strategy preferences while acknowledging that larger networks would further strengthen statistical inferences about the relationships between environmental conditions and optimal preprocessing approaches.

The focus on $PM_{2.5}$ reflects its critical importance for public health assessment and regulatory compliance. While this concentration provides a solid foundation for method development, extension to other atmospheric pollutants would benefit from parameter-specific validation, particularly for species with distinct temporal dynamics such as photochemical oxidants or traffic-related pollutants. The demonstrated cross-regional transferability between Pavlodar and Beijing suggests promising potential for broader geographic application, though validation in diverse climatic conditions would strengthen confidence in global applicability.

The 15-month study period captured multiple seasonal cycles and diverse pollution episodes, providing a robust foundation for method evaluation. This temporal coverage encompasses the typical range of atmospheric conditions encountered in operational monitoring, though longer-term studies would provide additional insights into method stability across inter-annual variability and extreme pollution events.

Methodological considerations inherent to the synthetic gap approach are addressed through systematic controls. Gap patterns reflect diverse failure scenarios commonly encountered in operational networks, while context availability assumptions align with realistic deployment scenarios where complete network failures are uncommon. The temporal splitting methodology preserves chronological dependencies, ensuring that evaluation reflects realistic prediction scenarios rather than retrospective interpolation.

The computational requirements of DynamicSeq2SeqXGB represent a practical consideration for implementation planning. While substantially more efficient than deep learning alternatives, the XGBoost-based approach requires greater processing resources than classical interpolation methods. This trade-off between accuracy and computational efficiency aligns with broader trends toward sophisticated environmental data analysis and can be managed through appropriate system design.

The emphasis on MAE as the primary evaluation metric provides interpretable error assessment directly relevant to air quality applications. The complementary use of RMSE, $R^2$, and cross-regional validation strengthens confidence in the performance assessment, while the observed patterns remain consistent across multiple evaluation approaches.

These considerations collectively indicate that while the current study provides strong evidence for DynamicSeq2SeqXGB's effectiveness within environmental monitoring applications, implementation benefits from local validation and

adaptation to specific operational requirements. The method's demonstrated robustness across diverse data quality conditions and geographic settings provides a solid foundation for broader deployment in operational monitoring networks facing data reconstruction challenges.

### 5.5. Future research directions

The demonstrated station-specific optimization principles warrant extension across broader environmental monitoring contexts. Validation of DynamicSeq2SeqXGB for other atmospheric pollutants ($O_3$, $NO_2$, $SO_2$, CO) would establish whether the adaptive context processing mechanisms generalize beyond $PM_{2.5}$ temporal dynamics. Each parameter presents distinct challenges: $O_3$'s photochemical cycles, $NO_2$'s traffic-related patterns, and $SO_2$'s industrial emission profiles may require modified hyperparameter configurations or alternative context window strategies.

Geographic expansion represents the most immediate research priority. Testing across diverse climatic zones – tropical monsoon regions, arctic conditions, coastal marine environments – would determine whether the Beijing-Pavlodar transferability extends to fundamentally different atmospheric regimes. These studies should focus on regions with established monitoring networks suffering chronic data quality issues rather than well-maintained systems.

Multi-station imputation architectures could enhance reconstruction for severely compromised sites by incorporating spatial correlations between monitoring locations. However, this extension requires addressing substantial technical challenges: synchronizing multi-dimensional input processing, managing variable network topologies, and scaling computational requirements. The potential benefits must be weighed against increased architectural complexity and data requirements.

Long-term performance assessment under extended deployment conditions would address questions about model stability during extreme pollution episodes, seasonal transitions, and gradual changes in local emission patterns. Particular attention should focus on performance degradation over time and requirements for model retraining or adaptation to evolving environmental conditions.

Investigation of alternative ensemble architectures beyond XGBoost – including CatBoost, LightGBM, or hybrid approaches combining gradient boosting with neural sequence models – may yield computational or accuracy improvements. These comparisons should emphasize operational feasibility rather than marginal performance gains unsuitable for practical deployment.

The methodology's applicability to other environmental time series with similar sparsity challenges – hydrological monitoring, soil sensor networks, biodiversity tracking systems –represents a logical extension within the environmental domain. These applications share characteristics of irregular data availability, temporal dependencies, and the need for robust reconstruction under challenging field conditions.

These directions should proceed with systematic validation rather than assuming transferability, focusing on domains where the method's specific strengths – handling extreme sparsity, adaptive context processing, station-specific optimization – address documented challenges in existing approaches.

## 6. Conclusions

This study demonstrates DynamicSeq2SeqXGB as an effective method for $PM_{2.5}$ imputation in severely compromised environmental monitoring networks. Across all stations and data completeness scenarios, the method achieved improvements of up to 79.8% compared to classical imputation approaches, with an average gain of 48.8% over standard baselines. These performance margins remained stable even under severe sparsity (23.3% to 57.5% completeness), highlighting the robustness of the proposed framework. The investigation of compression strategies revealed that optimal preprocessing depends critically on station-specific characteristics rather than universal quality thresholds, with full and selective compression showing equal overall effectiveness (50% win rate each) but distinct station preferences. Cross-regional validation using Beijing data confirmed transferability beyond the original training environment, achieving $R^2 = 0.944$ and maintaining 68.1–79.8% improvement margins over baseline methods across different geographic and pollution conditions.

The results provide actionable station-level guidance: stations with high temporal variability and unstable pollution dynamics consistently benefit from selective compression, whereas stations with more stable temporal structure perform better with full compression.

The practical implications center on adaptive, context-aware preprocessing strategies that optimize performance based on local data characteristics rather than conventional quality-first approaches. The approach proves most valuable for monitoring networks facing chronic data quality challenges where sufficient temporal structure remains for machine learning approaches to extract meaningful patterns. While demonstrating clear advantages within its tested scope, the method should be considered a specialized tool for challenging monitoring scenarios rather than a universal replacement for classical interpolation approaches, with future applications requiring systematic validation under specific operational conditions and careful consideration of computational requirements.

Future work may focus on developing an automatic station-specific strategy selector based on temporal and statistical characteristics, as well as extending the framework toward real-time adaptive imputation for online monitoring systems.

## Supporting information

**S1 File. Extended figures and complete benchmarking results.** *Fig S1.* Monthly $PM_{2.5}$ data completeness across five air quality monitoring stations in Pavlodar, Kazakhstan (May 2024 – July 2025), showing significant heterogeneity across stations and justifying full compression strategy adoption. *Figs S2A-E.* Data completeness heatmaps for each monitoring station, illustrating temporal patterns of missing and valid hourly observations with monthly completeness percentages. *Fig S3.* Scatter plots of true versus predicted $PM_{2.5}$ concentrations for DynamicSeq2SeqXGB across all stations with color-coded gap lengths (5h-72h), demonstrating model performance variability by station and temporal scale. *Fig S4.* Beijing $PM_{2.5}$ dataset characteristics including monthly completeness and concentration distributions from Guanyuan station (2016) used for external validation. *Fig S5.* Complex degradation analysis showing original time series, artificially degraded data (50% completeness), synthetic gap distribution, and DynamicSeq2SeqXGB reconstruction performance on Beijing dataset. *Fig S6.* Quantitative benchmarking comparison of imputation methods showing Mean Absolute Error (MAE) and Coefficient of Determination ($R^2$) across all evaluated approaches. *Fig S7.* Summary of station-level optimal compression strategies across five gap lengths (5, 12, 24, 48, and 72 hours). *Table S1.* Complete benchmarking results for all imputation methods across stations and gap lengths using full compression strategy, including MAE, RMSE, $R^2$, and MAPE metrics with standard deviations. *Table S2.* Complete benchmarking results for all imputation methods using selective compression strategy, providing parallel performance comparison for quality-filtered datasets. *Table S3.* Pearson correlation matrix of station characteristics and compression strategy performance metrics with calculation methodologies and significance indicators. *Table S4.* Beijing dataset experimental setup detailing synthetic degradation parameters, data partitions, and validation configuration for external generalization assessment.
(DOCX)

**S2 File. Technical protocols for data preprocessing and model configuration.** *Technical Protocol 1:* Step-by-step description of the outlier detection and statistical filtering pipeline for air quality data. *Technical Protocol 2:* Final hyperparameter specifications, feature engineering setup, and training configuration for the DynamicSeq2SeqXGB model.
(DOCX)

## Acknowledgments

The authors would like to express their gratitude to all individuals and organizations who contributed to this research through technical support, data provision, or academic guidance. Special thanks to colleagues who provided constructive feedback during the preparation of this manuscript.

## Author contributions

**Conceptualization:** Ruslan Safarov.

**Data curation:** Zhanat Shomanova.

**Formal analysis:** Eldar Kopishev.

**Funding acquisition:** Zhanat Shomanova.

**Investigation:** Yuriy Nossenko, Zhuldyz Bexeitova.

**Methodology:** Yuriy Nossenko.

**Project administration:** Ruslan Safarov.

**Resources:** Eldar Kopishev.

**Software:** Emin Atasoy.

**Supervision:** Emin Atasoy.

**Validation:** Zhanat Shomanova.

**Visualization:** Emin Atasoy.

**Writing – original draft:** Yuriy Nossenko, Zhuldyz Bexeitova.

**Writing – review & editing:** Ruslan Safarov.

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
