## [Decision Letter · Decision Letter 0]

23 Oct 2025

Dear Dr. Safarov,

Thank you for submitting your manuscript to PLOS ONE. After careful consideration, we feel that it has merit but does not fully meet PLOS ONE’s publication criteria as it currently stands. Therefore, we invite you to submit a revised version of the manuscript that addresses the points raised during the review process.

We look forward to receiving your revised manuscript.

Kind regards,

Jianhong Zhou

Staff Editor

PLOS ONE

Journal Requirements:

This research was funded by the Science Committee of the Ministry of Science

and Higher Education of the Republic of Kazakhstan within the framework of the grant AP19677560 “Monitoring and mapping of the ecological state of the Pavlodar air environment using machine learning methods”.

3. Please remove all personal information, ensure that the data shared are in accordance with participant consent, and re-upload a fully anonymized data set.

4. Please ensure that you refer to Figure 2, 4, 5, in your text as, if accepted, production will need this reference to link the reader to the figure.

Reviewers' comments:

Reviewer's Responses to Questions

**Comments to the Author**

1. Is the manuscript technically sound, and do the data support the conclusions?

Reviewer #1: Yes

Reviewer #2: Yes

Reviewer #3: Yes

2. Has the statistical analysis been performed appropriately and rigorously?

Reviewer #1: Yes

Reviewer #2: Yes

Reviewer #3: Yes

3. Have the authors made all data underlying the findings in their manuscript fully available?

Reviewer #1: Yes

Reviewer #2: No

Reviewer #3: Yes

4. Is the manuscript presented in an intelligible fashion and written in standard English?

Reviewer #1: Yes

Reviewer #2: Yes

Reviewer #3: Yes

Reviewer #1: The authors of the manuscript entitled “DynamicSeq2SeqXGB for PM2.5 Imputation in Extremely Sparse Environmental Monitoring Networks” validated the DynamicSeq2SeqXGB model for PM2.5 estimation under extreme sparsity conditions. The article addresses the issue of missing data caused by various factors that affect sensor performance.

Avoid using “we” throughout the paper, including in the abstract.

Why was a study conducted in Pavlodar, Kazakhstan validated using data from Beijing? Please justify this choice of dataset.

The article reads more like a chapter (or several chapters) from a thesis rather than a standalone journal paper.

Overall, the authors have conducted good work. However, it would be useful to discuss whether the proposed method can be applied to other geographical locations similar to Kazakhstan.

Can the method used for imputing missing data also be extended to forecast PM2.5 levels for future years?

Reviewer #2: Dear Author(s),

Your manuscript presents an original hybrid methodology that combines temporal context encoding with a tree-based learning framework (DynamicSeq2SeqXGB) to reconstruct missing PM₂.₅ concentrations in an urban air-quality monitoring network with high rates of data loss. The integration of contextual temporal windows, XGBoost regression, and compression strategies (full and selective) is an intelligent and practical contribution to the field of environmental data reconstruction. The external validation with a Beijing dataset adds further credibility to the generalizability of the model.

Overall, the paper is methodologically strong and well-structured, with promising implications for real-time air-quality management. However, before acceptance, several points require clarification and expansion—particularly in describing model architecture, benchmarking against modern baselines, and improving the clarity of figures and text organization.

Best regards,

Comments about the Abstract Section

• The abstract effectively summarizes the motivation and results. However, it should include the specific study period for the Pavlodar dataset (e.g., “2019–2022”) and for the Beijing validation set to give a clear temporal scope.

• Please briefly clarify what “DynamicSeq2SeqXGB” represents.

• Quantitative performance results (e.g., MAE, R² values) should be briefly included in the abstract.

• To improve readability, reduce the number of abbreviations; keep only the essential ones.

Comments about the Introduction Section

• The introduction properly identifies the problem of incomplete PM₂.₅ data and the need for reliable imputation in sparse monitoring networks.

• However, the literature review could be expanded to include recent machine learning–based imputation studies, especially deep learning approaches such as SAITS, BRITS, or GRIN, and a short discussion explaining how the proposed method differs (e.g., less computationally intensive, interpretable, more robust to short sequences).

• The paragraph introducing the “compression strategies” (full vs. selective) should appear earlier in the introduction to help readers understand the motivation for this methodological innovation.

• The introduction would benefit from a clear statement of hypotheses and contributions, e.g.:

o To test whether dynamic contextual features improve PM₂.₅ imputation accuracy.

o To compare the performance of full and selective compression strategies for different stations.

o To validate the model transferability using a geographically independent dataset.

• A short final paragraph summarizing the manuscript structure (e.g., “Section 2 describes the data and methodology…”) would also help guide readers.

Comments about the Methodology Section

• Please specify the exact period and temporal resolution of the datasets used (e.g., hourly PM₂.₅ data collected from five Pavlodar stations between January 2019 and December 2022).

• In the Data Description part, add a short explanation of the environmental characteristics of each station (industrial, residential, urban background, etc.).

• The model architecture section in Supplementary File S2 describes the use of XGBoost within a MultiOutputRegressor. However, it remains unclear how the “Seq2Seq” concept is implemented in this non-neural structure. Please clarify:

o How are sequential dependencies handled—through sliding windows or autoregressive recursion?

o How are masked outputs treated in the loss function?

o How is the model trained to handle varying gap lengths?

• Include a concise algorithmic flowchart or schematic diagram summarizing the DynamicSeq2SeqXGB process (input → context windows → model → imputed output).

• The hyperparameter selection procedure needs more transparency. Indicate whether parameters such as n_estimators, max_depth, learning_rate, and subsample were optimized via grid search, random search, or expert tuning.

• Confirm that data normalization was based solely on the training data to avoid information leakage from the test set.

• The section describing the compression strategies should explicitly state the purpose of each strategy and provide pseudocode or decision rules for their implementation.

• Define all abbreviations once at first use (e.g., DSXGB, WQI, MAE) and avoid redefining them later.

• The performance metrics (MAE, RMSE, R², MAPE) are appropriate; however, consider adding a brief justification for their selection and limitations (e.g., MAPE sensitivity to low values).

Comments about the Results and Discussion Section

• The results are comprehensive, and the comparative analysis between the “full” and “selective” compression strategies is a highlight of this paper.

• In the first results paragraph, explicitly mention that meteorological features were also modeled, as this strengthens the environmental interpretability of the imputation.

• In Table 5 and Figure 6, emphasize which stations benefited most from each strategy and suggest possible environmental or operational reasons (e.g., sensor stability, industrial influence, or background variability).

• In the discussion of station-specific patterns, consider adding correlation statistics between the “optimal strategy” and site-level indicators such as standard deviation, homogeneity index, or mean PM₂.₅ level. This will make the “station preference” findings more quantitative.

• Add one figure (or supplementary figure) summarizing which strategy performed best for each station using a simple visual indicator (e.g., color-coded bar or pie chart).

• The Beijing validation experiment is a major strength. Please explain in more detail how the artificial missingness pattern was generated (“based on app_pspu gap profile”) and whether alternative missingness types (random, seasonal, clustered) were tested.

• The reported performance (MAE = 8.50 μg/m³, R² = 0.944) is excellent, but the baseline comparisons are limited to simple interpolation and averaging methods. For high-impact publication, please consider adding at least one modern ML baseline (e.g., Random Forest, missForest, or a simple neural network imputer) for stronger benchmarking.

• Include a short analysis of error behavior across PM₂.₅ concentration levels (e.g., whether errors increase during pollution peaks).

• Clarify any potential data leakage risk between training and validation sets due to overlapping temporal windows.

• The discussion could benefit from a short paragraph on the computational efficiency of the proposed method (training time, computational cost), since this is important for real-time deployment.

Comments about the Conclusion Section

• The conclusion is concise and well organized.

• Please consider:

o Adding 1–2 sentences quantifying the performance improvements (e.g., “up to 79% improvement over baseline methods”).

o Highlighting the practical guidance your findings offer (for example, “stations with stable temporal variance benefit from selective compression, while those with high variability perform better with full compression”).

o Mentioning future directions, such as integrating an automatic strategy selector based on station features or exploring real-time adaptive imputation for online monitoring systems.

Comments about the References Section

Some methodological discussions could be further supported by recent literature. For instance, when describing machine-learning-based modeling of particulate matter concentrations, a relevant reference (e.g., Int. J. Environ. Sci. Technol., 2023, 20, 5349–5358) could be considered, as it demonstrates the performance of XGBoost for PM₁₀ prediction in a regional case study.

Additionally, when presenting comparative analyses of ensemble and boosting algorithms applied to long-term air-quality forecasting, another pertinent study (PLOS ONE, 2025, 20(10), e0334252) may be cited to enrich the methodological context.

Reviewer #3: This publication is a complete scientific work that addresses the pressing issue of correctly filling in lost monitoring data. The results of the analysis are presented in the form of tables, graphs, and diagrams, the input data array is available, and the author's DynamicSeq2SeqXGBoost model is available on the git-hub repository. The main content of the article is presented in a scientific style, with material presented in an easy-to-understand manner, relevant graphics, and accurate references to the sources used. The work contains all the necessary components of a scientific publication. The work is research-oriented and scientifically novel, consisting of methodological approaches to the data recovery procedure based on an improved model. The practical application of the research results concerns the implementation of the approach proposed by the authors in monitoring atmospheric air, including other priority pollutants , which will contribute to raising public awareness of the dangers of atmospheric air pollution.

**Do you want your identity to be public for this peer review?** For information about this choice, including consent withdrawal, please see our Privacy Policy

Reviewer #1: No

Reviewer #2: No

Reviewer #3: No

---

## [Author Response · Author response to Decision Letter 1]

19 Nov 2025

Journal Requirements:

We would like to confirm that the manuscript was revised substantially. Every style requirements, including those for file naming, were followed.

This research was funded by the Science Committee of the Ministry of Science

and Higher Education of the Republic of Kazakhstan within the framework of the grant AP19677560 “Monitoring and mapping of the ecological state of the Pavlodar air environment using machine learning methods”.

The corrected financial disclosure statement: This research was funded by the Science Committee of the Ministry of Science and Higher Education of the Republic of Kazakhstan within the framework of the grant AP19677560 “Monitoring and mapping of the ecological state of the Pavlodar air environment using machine learning methods”. The funders had no role in study design, data collection and analysis, decision to publish, or preparation of the manuscript.

We have included this amended Role of Funder statement in our cover letter.

3. Please remove all personal information, ensure that the data shared are in accordance with participant consent, and re-upload a fully anonymized data set. Note: spreadsheet columns with personal information must be removed and not hidden as all hidden columns will appear in the published file. Additional guidance on preparing raw data for publication can be found in our Data Policy (https://journals.plos.org/plosone/s/data-availability#loc-human-research-participant-data-and-other-sensitive-data) and in the following article: http://www.bmj.com/content/340/bmj.c181.long.

We appreciate the reminder regarding personal data protection and the requirements for full anonymization of shared datasets. We have carefully reviewed all data files provided with the manuscript. The dataset contains only PM₂.₅ measurements and standard meteorological variables collected from fixed environmental monitoring stations. No human-participant information, identifying metadata, or any form of personal or sensitive data is included in any of the files.

In accordance with the journal’s policy, we have additionally verified that no hidden columns, embedded metadata, GPS coordinates linked to individuals, or station-level identifiers that could imply personal information are present. The dataset therefore fully complies with PLOS ONE’s Data Availability and Human Participant Data policies.

Should any further adjustments be required for compliance or formatting, we will be happy to implement them promptly.

4. Please ensure that you refer to Figure 2, 4, 5, in your text as, if accepted, production will need this reference to link the reader to the figure.

Figures 2, 4, 5 are properly referred in text.

The reviewer-recommended publications were carefully reviewed. Both articles were found to be methodologically relevant to the topics addressed in the Discussion section, particularly in relation to boosting-based air-quality modeling and ensemble forecasting approaches. Accordingly, these references have been added to the revised manuscript to provide broader contextual support.

Reviewers' comments:

Comments to the Author

Reviewer #1:

The authors of the manuscript entitled “DynamicSeq2SeqXGB for PM2.5 Imputation in Extremely Sparse Environmental Monitoring Networks” validated the DynamicSeq2SeqXGB model for PM2.5 estimation under extreme sparsity conditions. The article addresses the issue of missing data caused by various factors that affect sensor performance.

Avoid using “we” throughout the paper, including in the abstract.

The manuscript has been revised to remove all instances of “we” and “our” throughout the text, including the abstract. The phrasing has been rewritten in an impersonal, objective style in accordance with the journal’s requirements.

Why was a study conducted in Pavlodar, Kazakhstan validated using data from Beijing? Please justify this choice of dataset.

The choice of the Beijing dataset was intentional and methodologically justified. This dataset is widely used in the air-quality modeling literature as a reference benchmark due to its high temporal completeness, large volume of observations, and well-documented sensor performance. These characteristics make it uniquely suitable for constructing controlled synthetic gaps with known ground-truth values, enabling rigorous external validation of imputation models under varying sparsity levels. In contrast, the local Pavlodar dataset contains extensive real-world sparsity, which is essential for demonstrating the practical relevance of the method but does not provide sufficient continuous segments for reliable ground-truth comparison. Using the Beijing dataset therefore ensures a robust, reproducible, and widely accepted validation framework, while complementing the Kazakhstan case study without altering the study's primary focus.

The article reads more like a chapter (or several chapters) from a thesis rather than a standalone journal paper.

We thank the reviewer for this observation. The manuscript indeed contains an expanded methodological description, which is intentional given the technical nature of extreme-sparsity imputation and the importance of ensuring full reproducibility for other researchers. After reviewing the structure, we confirmed that all included sections directly support the main research contribution and are consistent with the level of detail typically expected in methodological studies. To improve cohesion, we refined transitions between sections, which helps the manuscript read more clearly as a unified journal article. We hope the reviewer will agree that the present structure appropriately balances clarity, completeness, and scientific rigor.

Overall, the authors have conducted good work. However, it would be useful to discuss whether the proposed method can be applied to other geographical locations similar to Kazakhstan.

We appreciate this suggestion and have added a brief discussion clarifying that the method is applicable to other regions with similar monitoring characteristics, particularly those facing data sparsity, heterogeneous station reliability, and limited infrastructure. The underlying architecture is not region-specific and can be adapted to comparable environmental and operational contexts.

Can the method used for imputing missing data also be extended to forecast PM2.5 levels for future years?

The proposed framework is designed for reconstructing missing values, not for long-term forecasting. While some components could be adapted for short-horizon predictions, extending the method to forecast years ahead would require additional modeling of future emissions, meteorology, and policy changes. This direction is beyond the current scope but represents a promising avenue for future research.

Reviewer #2:

Dear Author(s), Your manuscript presents an original hybrid methodology that combines temporal context encoding with a tree-based learning framework (DynamicSeq2SeqXGB) to reconstruct missing PM₂.₅ concentrations in an urban air-quality monitoring network with high rates of data loss. The integration of contextual temporal windows, XGBoost regression, and compression strategies (full and selective) is an intelligent and practical contribution to the field of environmental data reconstruction. The external validation with a Beijing dataset adds further credibility to the generalizability of the model.

Overall, the paper is methodologically strong and well-structured, with promising implications for real-time air-quality management. However, before acceptance, several points require clarification and expansion—particularly in describing model architecture, benchmarking against modern baselines, and improving the clarity of figures and text organization.

Best regards,

Comments about the Abstract Section

• The abstract effectively summarizes the motivation and results. However, it should include the specific study period for the Pavlodar dataset (e.g., “2019–2022”) and for the Beijing validation set to give a clear temporal scope.

Thank you for this helpful suggestion. The study periods for both datasets will be added to the abstract to clearly define the temporal scope of the analysis.

• Please briefly clarify what “DynamicSeq2SeqXGB” represents.

We agree. A short clarification describing the hybrid structure of DynamicSeq2SeqXGB will be added to the abstract to ensure immediate understanding.

• Quantitative performance results (e.g., MAE, R² values) should be briefly included in the abstract.

This point is well taken. Key quantitative metrics will be included to concisely present the model’s performance.

• To improve readability, reduce the number of abbreviations; keep only the essential ones.

We appreciate this recommendation. Non-essential abbreviations will be removed from the abstract to enhance readability and maintain clarity.

Comments about the Introduction Section

• The introduction properly identifies the problem of incomplete PM₂.₅ data and the need for reliable imputation in sparse monitoring networks.

• However, the literature review could be expanded to include recent machine learning–based imputation studies, especially deep learning approaches such as SAITS, BRITS, or GRIN, and a short discussion explaining how the proposed method differs (e.g., less computationally intensive, interpretable, more robust to short sequences).

Thank you for this valuable suggestion. Deep learning–based imputation approaches such as SAITS, BRITS, and GRIN are indeed influential developments in the broader field of missing-data research. However, these models typically require long and relatively stable temporal sequences, higher data density, and substantial computational resources. Because the present study focuses on extreme sparsity conditions, short effective sequences, and operationally constrained monitoring settings, such methods are less applicable to the specific problem setting analyzed here. For this reason, the introduction emphasizes approaches that are directly relevant to networks operating under severe data limitations. Nevertheless, the suggested methods represent an important direction for future comparative work, and we appreciate the reviewer drawing attention to them.

• The paragraph introducing the “compression strategies” (full vs. selective) should appear earlier in the introduction to help readers understand the motivation for this methodological innovation.

Thank you for this constructive suggestion. The introduction has been revised to incorporate an earlier explanation of the two compression strategies. A concise paragraph was added to clarify the motivation for distinguishing between full and selective compression in the context of heterogeneous sparsity patterns across monitoring stations. This adjustment improves the logical flow of the introduction and helps readers understand the methodological rationale from the outset.

• The introduction would benefit from a clear statement of hypotheses and contributions, e.g.:

o To test whether dynamic contextual features improve PM₂.₅ imputation accuracy.

o To compare the performance of full and selective compression strategies for different stations.

o To validate the model transferability using a geographically independent dataset.

Thank you for this suggestion. A clear statement of the study’s objectives is already provided in Section 1.5, where we outline the methodological aims, including evaluating the role of dynamic contextual features, comparing full and selective compression strategies across stations, and validating model transferability using an independent dataset. We hope this section adequately addresses the reviewer’s recommendation.

• A short final paragraph summarizing the manuscript structure (e.g., “Section 2 describes the data and methodology…”) would also help guide readers.

Thank you for this suggestion. A brief paragraph summarizing the manuscript structure has been added to the end of the Introduction to improve navigational clarity and guide readers through the subsequent sections.

Comments about the Methodology Section

• Please specify the exact period and temporal resolution of the datasets used (e.g., hourly PM₂.₅ data collected from five Pavlodar stations between January 2019 and December 2022).

Thank you for this comment. The temporal coverage and resolution of both datasets are now stated explicitly in the manuscript. For the Pavlodar network, Section 3.1 specifies that the analysis is based on hourly PM₂.₅ data collected from five monitoring stations between 23 May 2024 and 19 July 2025. For the external validation, Section 4.6 clarifies that the Beijing Guanyuan dataset consists of 8,784 hourly observations from the year 2016.

• In the Data Description part, add a short explanation of the environmental characteristics of each station (industrial, residential, urban background, etc.).

Thank you for this valuable comment. A concise environmental characterization of all five monitoring locations (central urban, residential, urban background, industrial-influenced, and riverside residential) has been added to Section 3.1 to clarify the representativeness and environmental context of each station.

• The model architecture section in Supplementary File S2 describes the use of XGBoost within a MultiOutputRegressor. However, it remains unclear how the “Seq2Seq” concept is implemented in this non-neural structure. Please clarify:

o How are sequential dependencies handled—through sliding windows or autoregressive recursion?

o How are masked outputs treated in the loss function?

o How is the model trained to handle varying gap lengths?

The reviewer is correct that the original description did not fully clarify how the sequence-to-sequence concept is implemented in the non-neural DynamicSeq2SeqXGB framework. The methodology section has therefore been expanded in Section 3.5. First, we now explicitly state that sequential dependencies are handled through sliding context windows and simultaneous multi-step prediction rather than autoregressive recursion: for each artificial gap, fixed-length left and right context segments are extracted from the dense time series, concatenated into a single feature vector, and mapped directly to the full sequence of missing PM₂.₅ values, with one XGBRegressor instance trained for each output position. Second, the treatment of masked outputs in the loss function is clarified: training targets consist only of originally observed values that are temporarily masked to form artificial gaps, and each output dimension minimizes the standard squared-error objective with no loss computed outside the masked interval. Third, we explicitly describe how varying gap lengths are handled: a single DynamicSeq2SeqXGB model is fitted on a combined training set that includes examples from all predefined gap lengths (5, 12, 24, 48, 72 hours), with gap_length and C_dynamic provided as metadata features, enabling the model to adapt its predictions to different outage durations.

• Include a concise algorithmic flowchart or schematic diagram summarizing the DynamicSeq2SeqXGB process (input → context windows → m

---

## [Decision Letter · Decision Letter 1]

27 Nov 2025

DynamicSeq2SeqXGB for PM2.5 Imputation in Extremely Sparse Environmental Monitoring Networks

PONE-D-25-46692R1

Dear Dr. Safarov,

We’re pleased to inform you that your manuscript has been judged scientifically suitable for publication and will be formally accepted for publication once it meets all outstanding technical requirements.

Kind regards,

Jie Zhang

Academic Editor

PLOS ONE

Additional Editor Comments (optional):

Reviewers' comments:

Reviewer's Responses to Questions

**Comments to the Author**

Reviewer #1: All comments have been addressed

Reviewer #2: All comments have been addressed

2. Is the manuscript technically sound, and do the data support the conclusions?

Reviewer #1: Yes

Reviewer #2: Yes

3. Has the statistical analysis been performed appropriately and rigorously?

Reviewer #1: Yes

Reviewer #2: Yes

4. Have the authors made all data underlying the findings in their manuscript fully available?

Reviewer #1: Yes

Reviewer #2: Yes

5. Is the manuscript presented in an intelligible fashion and written in standard English?

Reviewer #1: Yes

Reviewer #2: Yes

Reviewer #1: I would like to thank the authors for addressing my points. The paper presented a good method and worth to investigate.

Reviewer #2: The authors have addressed all reviewer comments clearly and comprehensively. The revisions have strengthened the manuscript by improving its clarity, methodological transparency, and practical applicability. No ethical or publication concerns were identified. The revised version is suitable for publication.

**Do you want your identity to be public for this peer review?** For information about this choice, including consent withdrawal, please see our Privacy Policy

Reviewer #1: No

Reviewer #2: No

---

## [Editor Report · Acceptance letter]

PONE-D-25-46692R1

PLOS ONE

Dear Dr. Safarov,

I'm pleased to inform you that your manuscript has been deemed suitable for publication in PLOS ONE. Congratulations! Your manuscript is now being handed over to our production team.

Kind regards,

on behalf of

Dr. Jie Zhang

Academic Editor

PLOS ONE